# INTERNAL PLANNING IN LANGUAGE MODELS: CHARACTERIZING HORIZON AND BRANCH AWARENESS

**Muhammed Ustaomeroglu**\*, **Baris Askin**\*, **Gauri Joshi, Carlee Joe-Wong, Guannan Qu**
Department of Electrical and Computer Engineering
Carnegie Mellon University
Pittsburgh, PA 15213, USA
`{mustaome,baskin,gaurij,cjoewong,gqu}@andrew.cmu.edu`

## ABSTRACT

The extent to which decoder-only language models (LMs) engage in planning, that is, organizing intermediate computations to support coherent long-range generation, remains an important question, with implications for interpretability, reliability, and principled model design. Planning involves structuring computations over long horizons, and considering multiple possible continuations, but how far transformer-based LMs exhibit them without external scaffolds, e.g., chain-of-thought prompting, is unclear. We address these questions by analyzing the hidden states at the core of transformer computations, which capture intermediate results and act as carriers of information. Since these hidden representations are redundant and encumbered with fine-grained details, we develop a pipeline based on vector-quantized variational autoencoders that compresses them into compact summary codes. These codes enable measuring mutual information and analyzing the computational structure of the underlying model behavior. Using this framework, we study planning in LMs across synthetic grammar, path-finding tasks, and natural language datasets, focusing on two planning properties: (i) the planning horizon of pre-output computations, and (ii) the extent to which the model considers alternative valid continuations. As a separate downstream use of the same pipeline, we also analyze how decision-relevant information is distributed across layers and earlier prefix blocks when producing next-token predictions. Together, these analyses advance our understanding of planning in LMs and provide a general-purpose pipeline for inspecting internal model dynamics. Our results reveal that the effective planning horizon is task-dependent, that models implicitly preserve information about unused correct continuations, and that predictions draw most on recent computations, though earlier blocks remain informative.

## 1 INTRODUCTION

Language models (LMs) have advanced so rapidly that they now engage in open-ended conversation, write functional code, and even solve challenging mathematical problems (Brown, 2020; Achiam et al., 2023; Grattafiori et al., 2024; Touvron et al., 2023). Beyond these raw capabilities, scaffolding and augmentation techniques further enhance reasoning and planning (Wei et al., 2022; Wang et al., 2024c; Sel et al., 2025; Yao et al., 2023; Besta et al., 2024). In parallel, hybrid systems integrating LMs with symbolic planners or external tools have achieved state-of-the-art performance in embodied and tool-use domains (Zhao et al., 2023; Wang et al., 2024a; Shen et al., 2023).

All the above success suggests that LMs have a certain capability of "planning ahead," similar to what humans do in generating coherent long speeches and long-horizon problem solving. Yet, the typical training method for LMs, next token prediction, focuses only on predicting the next token, which seemingly suggests LMs are myopic. Motivated by this contrast, this paper seeks to understand the planning behavior of LMs and how it is affected by next-token prediction (NTP) vs. multi-token prediction (MTP) in training. As "planning behavior" is a broad term that has many facets, we further narrow our focus to the following two canonical aspects of planning. First, good

---

\*Equal contribution

planners are *forward-looking*. They structure intermediate computations over a receding horizon so that near-term actions serve longer-term objectives, an idea made explicit in model-predictive control (MPC) and world-model–based approaches (Morari & Lee, 1999; Mayne et al., 2000; Ha & Schmidhuber, 2018; LeCun, 2022). Second, good planners are *branch-aware*, before committing, they keep multiple plausible futures "alive," comparing candidates rather than greedily following a single line, an ability central in classical MPC and world-model–based control methods (Morari & Lee, 1999; Mayne et al., 2000; Hafner et al., 2019), and closely tied to robustness under paraphrase and logically related prompts (Lin et al., 2025; Ahn & Yin, 2025; Saxena et al., 2024) and to the success of search-based scaffolds such as Tree- and Graph-of-Thoughts (Yao et al., 2023; Besta et al., 2024). Given the above aspects about planning, this paper seeks to understand,

*To what extent are LMs forward-looking and branch-aware?*
*How does the training method (NTP vs. MTP) affect these qualities?*

We approach the above questions by studying the *internal states and computations* of an LM, as recent work shows that transformers internally encode rich, high-level abstractions such as belief states, game configurations, and world models that extend far beyond their immediate outputs, and that these abstractions can be partially extracted (Shai et al., 2024; Li et al., 2023; Pal et al., 2023; Richens et al., 2025). This makes the internal representations of LMs a natural locus for investigating how they plan and reason. While not aimed at these specific questions, existing methods have explored internal computations of models, e.g. using probing classifiers to test for linguistic features in hidden states or mechanistic analyses to identify circuits and disentangled features within transformers (Hewitt & Liang, 2019; Elhage et al., 2021; Bricken et al., 2023). However, these only indirectly relate to our questions, and despite important progress, these strategies have clear limitations. Circuit discovery requires heavy manual engineering (Elhage et al., 2021; Wang et al., 2023; Chan et al., 2022; Meng et al., 2022), while probing risks conflating genuine representations with probe artifacts (Hewitt & Liang, 2019; Voita & Titov, 2020; Pimentel et al., 2020; Kunz & Kuhlmann, 2020; Kumar et al., 2022). These limitations necessitate new approaches towards understanding the internal computations that are automated, scalable, and less susceptible to probing's confounding of representations learned by the probe with those in the model itself.

**Contribution 1:** We propose an information-theoretic framework to study how planning-related computations are organized inside LMs that is both automated (no manual circuit engineering) and free from confounding caused by learned probes. Specifically, to avoid probe-induced confounding, we compute mutual information (MI) between learned discrete representations that summarize internal states of the LM. By comparing these MI relationships, we characterize whether internal computations exhibit patterns consistent with forward-looking and branch-aware computation. For example, to understand the "forward-looking" characteristic, the MI between the internal states of the prefix and that of future generations can shed light on how much computations inside LMs plan ahead for future tokens. In contrast, probing aims to detect whether a piece of information is present in a LM hidden state by training an external model to predict the information from the hidden state, which may introduce additional representational power brought by the external model. The resulting prediction losses are often treated as informal proxies for "how much information" a hidden state contains but they depend on the marginal complexity and scale of the chosen target and can violate basic information-theoretic desiderata such as data processing. As we show in our experiments (App. D.2), such probe-based quantities can be strongly influenced by these nuisance factors and need not track true mutual information even in simple controlled settings. In contrast, computing MI between learned representations gives a confound-resistant metric of how much two variables share information, without introducing an additional supervised model whose capacity or objective may obscure the underlying LM computation. Moreover, unlike correlation, which is sensitive primarily to linear relationships under a chosen parameterization, MI captures arbitrary statistical dependence and is symmetric and invariant under invertible reparameterizations. To make the MI calculation *scalable* to potentially very high dimensional hidden state vectors, we employ a Vector-Quantized Variational Autoencoder (`VQ-VAE`) to map collections of block outputs into discrete codes that serve as coarse summaries of internal states (Van Den Oord et al., 2017). We use `VQ-VAE` as a practical compressor with a discrete codebook and transformer encoder, enabling MI estimation over variable-sized blocks; our validation study (App. A.4) supports this choice. We calculate MI between the coarse summaries instead of the raw hidden states. This step is crucial: fine-grained activations are high-dimensional and redundant, making a detailed direct analysis infeasible, while the compressed codes capture the salient distinctions necessary for comparing computations across layers, positions, and contexts. Beyond these planning analyses, the same pipeline also supports

a practical diagnostic: localizing where next-token decision information resides across layers and earlier prefix blocks. While our framework is general and could be applied to other applications in deep learning, in this work, we use it to analyze planning in transformer models, and we additionally demonstrate the localization diagnostic as a method application.

**Contribution 2: Understanding the planning of LMs.** Using the information-theoretic framework, we analyze LMs' ability to be (i) forward-looking, (ii) branch-aware, and (iii) as an application of our framework, how layers and earlier prefix blocks contain next token information, across a symbolic task, a structured reasoning problem, and natural text (§ 3):

- **Horizon of the Plan** (§ 3.1): how far ahead a model plans before producing its next token.
- **Branching in the Plan** (§ 3.2): to what extent a LM internally considers alternative responses.
- **Diagnostic of the information in the computational history** (§ 3.3): where decision-relevant information about the next token is concentrated across layers and earlier prefix blocks.

Our results show that planning is task-contingent and weakly modulated by the training objective loss, i.e., MTP versus NTP. When the task demands a longer horizon, the LM's pre-output states retain information about tokens beyond the immediate next step. In contrast, in locally syntactic settings, this dependence concentrates near the next token. Training with MTP loss modestly reduces purely myopic behavior. Internally, the model encodes alternatives to the produced answer, and the strength of this branching awareness of the plan varies with task difficulty and model quality. Finally, next-token decisions draw most strongly on the last layers and the most recent token indices.

**Related work.** Despite the recent advances in LM planning abilities (Wang et al., 2024c; Sel et al., 2025; Wei et al., 2022; Yao et al., 2023; Besta et al., 2024; Zhao et al., 2023; Wang et al., 2024a; Shen et al., 2023), recent studies highlight that significant challenges remain, and current approaches to LM planning still fall short of fully addressing complex reasoning and decision-making tasks. For example, Lin et al. (2025) show that models can produce conflicting answers under logically related prompts despite local plausibility, which shows lack of long-horizon planning; Ahn & Yin (2025); Saxena et al. (2024); Momennejad et al. (2023); Wang et al. (2024b) highlight inconsistencies in the model's outputs and struggles with planning tasks. Taken together, these findings underscore that understanding whether and how planning arises in LMs is not only an open empirical challenge, but also central to both their interpretability and the principled design of future model architectures.

To interpret LMs, some approaches treat the model as a black box and design tasks or benchmarks to gauge reasoning, robustness, or generalization abilities at a behavioral level (Srivastava et al., 2023; Liang et al., 2022). While such evaluations provide useful insights, they miss several perspectives that can be gained by examining the internal mechanisms of the model. In contrast, mechanistic interpretability seeks to reverse-engineer transformer computations into human-understandable parts, treating the residual stream as the main information pathway and attention heads as separate components that pass information along (Elhage et al., 2021; Olsson, 2022; Cunningham et al., 2023; Bricken et al., 2023; Hewitt & Liang, 2019; Dunefsky et al., 2024; Lindsey et al., 2025). However, circuit discovery requires significant manual engineering (Wang et al., 2023; Chan et al., 2022; Meng et al., 2022). Beyond empirical tools, mathematically grounded perspectives also shed light on transformer and LLM interpretability (Liu et al., 2023; Ahn et al., 2023; Ustaomeroglu & Qu, 2025; Gao et al., 2024), yet these approaches are often criticized for lacking a one-to-one correspondence with experimental results, since their proofs typically rely on strong assumptions that may not hold for the highly non-smooth LM architectures. In contrast to them, our method (§ 2) enables the study of LMs both behaviorally and structurally, by examining their internal mechanisms, without the need for labor-intensive circuit discovery or strong assumptions on the LM. Still, due to its reliance on information-theoretic tools, our approach can only capture aggregate phenomena, providing insights in an average sense rather than interpreting individual input prompts.

A different line of interpretability work views LM hidden states as structured representations that can be "probed"-probing refers to training lightweight models, often linear classifiers, to read out specific information from hidden states in order to test what the model represents internally. Using probing, researchers have shown that transformer hidden states encode structured belief-state and world-model–like information (Shai et al., 2024; Gurnee & Tegmark, 2024; Hazineh et al., 2023). Other works demonstrate that a single hidden state can carry information about multiple future tokens (Pal et al., 2023; Wu et al.) and that probing can reveal the underlying algorithms LLMs use to solve tasks (Allen-Zhu, 2024). Although there are some mathematical foundations for them (Xu

et al.), standard accuracy-based probing has been criticized for combining what the probe can learn with what the representation actually encodes, making the results sensitive to probe capacity, data size, and hyperparameters (Hewitt & Liang, 2019; Pimentel et al., 2020). Further, high probe scores often come from exploiting superficial linear context cues rather than genuine structural knowledge (Kunz & Kuhlmann, 2020): probing can even reveal features that a model does not use for its task (Ravichander et al., 2021; Kumar et al., 2022). Consequently, some critiques motivate adopting information-theoretic lenses (Voita & Titov, 2020; Diego-Simón et al., 2025). Similar to some of these information-theoretic approaches (Tishby & Zaslavsky, 2015; Shwartz-Ziv & Tishby, 2017; Skean et al., 2025; Voita et al., 2019), our method makes use of information theory. For a more detailed discussion of related work, please refer to App. C.

## 2 METHOD

Given a prefix sequence of length $T$, $\mathbf{x}_{1:T} = [x_1, x_2, \ldots, x_T]$, the decoder-only LM, $\mathcal{M}$, generates subsequent tokens $\{\widehat{x}_{T+\tau} \mid \tau = 1, 2, \ldots\}$ autoregressively. We denote the hidden activation at position $t$ after the $\ell^{\text{th}}$ transformer layer by $h_t^\ell \in \mathbb{R}^d$ where $d$ is the hidden dimension of the model.

In our analysis, we treat the outputs $h_t^\ell$ of individual transformer blocks as the fundamental units of computation[1]. We posit that a block's output is informative about the computation performed within that block (Pal et al., 2023; Shai et al., 2024; Li et al., 2023; Elhage et al., 2021; Lindsey et al., 2025). We use them as a proxy for the model's internal processing since each block's computation contributes to the overall function of the model. To analyze how planning emerges in the model's internal dynamics, we examine collections of layer–token indices $\mathcal{S}$ and aggregate the corresponding block outputs as $G_{\mathcal{S}} = \{h_t^\ell \mid (\ell, t) \in \mathcal{S}\}$, a representation for the computations within its block. While $G_{\mathcal{S}}$ provides a fine-grained representation of block computations, it often contains unnecessary details for our planning analysis. Hidden states may encode small variations due to token positions, normalization shifts, or attention noise, factors that are critical for exact output reconstruction but irrelevant for studying planning structure. Moreover, MI estimates between such high-dimensional vectors have high variance and are hard to find. To address both issues, we construct coarse, discrete representations of $G_{\mathcal{S}}$. These abstractions preserve the structural dependencies required for our analysis while filtering out irrelevant micro-level variability, yielding more stable and interpretable MI estimates. We depict the proposed method in Fig. 1. In the first step, we train a model that maps each collection $G_{\mathcal{S}}$ to a coarse discrete representation $Z_{\mathcal{S}}$. Using the trained model, we then obtain dataset-wide codes for the target hidden states and use these to estimate mutual information between the coarse representations of LM's internal computations.

**Measuring mutual information between coarse representations.** To investigate how planning is realized within the model, we use MI as a principled way to quantify relationships between the coarse representations of different computations, corresponding to Steps 2 and 3 of Fig. 1. Suppose we focus on two hidden state blocks, $G_A$ and $G_B$, which are random variables depending on the data distribution and the trained model parameters. We denote their coarse, discrete representations by $Z_A \in \mathcal{Z}_A$ and $Z_B \in \mathcal{Z}_B$ where $\mathcal{Z}_A$ and $\mathcal{Z}_B$ are the finite alphabets of possible codes. For realizations $z_a \in \mathcal{Z}_A$ and $z_b \in \mathcal{Z}_B$, $p(z_a, z_b)$, $p(z_a)$, and $p(z_b)$ refer to the joint probability of $Z_A$ and $Z_B$, and the marginal probabilities of $Z_A$ and $Z_B$, respectively. Mutual information, which measures how much knowing $Z_A$ reduces uncertainty about $Z_B$ (and vice versa), is defined as,

$$I(Z_A; Z_B) = \sum_{z_a \in \mathcal{Z}_A} \sum_{z_b \in \mathcal{Z}_B} p(z_a, z_b) \log \frac{p(z_a, z_b)}{p(z_a) \, p(z_b)}. \tag{1}$$

Since the true data distribution is inaccessible, we approximate by replacing $p(z_a)$, $p(z_b)$, and $p(z_a, z_b)$ in Eq. 1 with their empirical estimates from the datasets, obtaining by counting code pairs as in Step 2 of Fig. 1. We analyze the finite-sample error of this count-based MI estimator (and a bias-corrected variant) in App. E. We do not interpret MI as a direct measure of planning; rather, we use MI comparatively. We further define normalized mutual information (nMI) metric as

$$\text{nMI}(Z_A; Z_B) = \frac{I(Z_A; Z_B)}{\mathcal{I}_{\max}}. \tag{2}$$

where $\mathcal{I}_{\max}$ denotes the maximum value across the set of MI calculations performed in the same experimental analysis for comparison. For our purpose of LM planning investigation, the nMI metric

---

[1] We use the terms "hidden state" and "(hidden) activation" interchangeably to refer to these outputs.

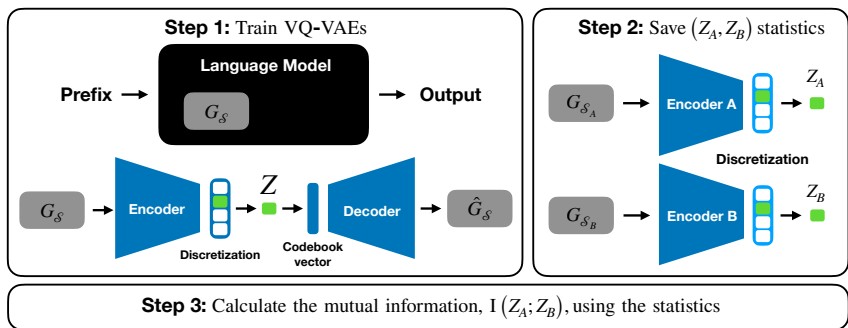

Figure 1: **The proposed method. Step 1 (training):** For a frozen LM $\mathcal{M}$, hidden states from selected transformer blocks $G_S$ are passed through a VQ-VAE encoder, which maps each block to a latent vector and then to a discrete codebook index $Z_S \in [K]$, providing coarse summaries of internal computations. **Step 2 (analysis):** For two sets of hidden-state blocks, $G_{S_A}$ and $G_{S_B}$, we apply the trained encoder and codebook to the dataset to obtain discrete variables $Z_A$ and $Z_B$ and collect their empirical co-occurrence counts. **Step 3:** Using these statistics, we estimate joint and marginal distributions $p(z_a, z_b)$, $p(z_a)$, and $p(z_b)$ to compute mutual information $I(Z_A; Z_B)$ (and its normalized variant). Then, we analyze how information is shared between different components of the model's computation.

is sufficient since our analyses depend on relative, rather than absolute, comparisons between sets of computations. Accordingly, all claims are about within-setting trends, not about nMI as an absolute planning score. Absolute values can be misleading for three reasons: (i) the estimated MI of coarse representations is bounded above by the logarithm of the alphabet size, making it inherently dependent on the experimental choice; (ii) any coarse-graining necessarily discards some information, which is advantageous for our setting as discussed but removes any meaning of absolute MI values related to the original hidden states; and (iii) answering our three research questions (§ 3) requires only relative comparisons of MI instead of relying on absolute values.

**Acquiring coarse representations.** To estimate MI between the high-level representations of hidden state blocks, $G_S$, we need to find a mapping from $G_S$ to the discrete codes, as depicted in Step 1 of Fig. 1. The size of target index set $S$ and its corresponding activation outputs $G$ may vary depending on the analysis. For instance, when examining all block outputs across the prefix tokens, $S$ spans every layer and grows with the prefix length $T$. Since neural networks are effective at learning representations, we employ an encoder-decoder neural network architecture: the encoder maps the potentially long and variable-sized $G_S$ to a compact representation, and the decoder aims to recover $G_S$ from it. Leveraging the effectiveness in handling variable-length inputs and learning rich representations (Lin et al., 2022), we adopt transformer-based encoder and decoder models. Specifically, we employ Vector-Quantized Variational Autoencoder (VQ-VAEs; Van Den Oord et al. 2017). VQ-VAE is a framework consisting of an encoder-decoder network with a trainable embedding codebook between them to discretize the encoder's output.

Concretely, for each family of index sets $S$ used in our analyses, we train a separate single VQ-VAE consisting of an encoder $E$, a codebook $\{e_k\}_{k=1}^{K} \subset \mathbb{R}^{d_e}$, and a decoder $D$. Given a block $G_S$, the encoder produces a latent vector $r_S = E(G_S) \in \mathbb{R}^{d_e}$. We then quantize by nearest neighbor in the codebook,

$$k^\star = \arg\min_{k \in [K]} \|r_S - e_k\|_2^2, \qquad Z_S \equiv k^\star, \qquad \tilde{r}_S \equiv e_{k^\star},$$

and reconstruct $\widehat{G}_S = D(\tilde{r}_S)$. During the training of VQ-VAE model, gradients are propagated through the quantization step using the straight-through estimator (Van Den Oord et al., 2017). The overall training objective combines reconstruction, standard vector-quantization, and two regularization terms that promote informative and well-used codes:

$$\mathcal{L} = \mathcal{L}_{\text{rec}} + \lambda_q \mathcal{L}_{\text{vq}} + \lambda_{\cos} \mathcal{L}_{\cos} + \lambda_{\text{ent}} \mathcal{L}_{\text{ent}}. \tag{3}$$

In Eq. 3, $\mathcal{L}_{\text{rec}}$ is a reconstruction loss that encourages $\widehat{G}_S$ to match $G_S$, $\mathcal{L}_{\text{vq}}$ is the quantization and commitment loss as in Van Den Oord et al. (2017) tying encoder outputs to discrete codes. In

addition to vanilla VQ-VAE's training objective we introduce $\mathcal{L}_{\text{cos}}$, a cosine-similarity penalty that pushes codebook embeddings $\{e_k\}$ to be diverse, and $\mathcal{L}_{\text{ent}}$, an entropy term that discourages collapse to a small subset of codes. These losses encourage representations assigned to different codes to diverge so that they are not only compact but also discriminative.[2] This enables embeddings to spread out in the latent space and ensures that codes capture distinct features of the data making them informative and discriminative. Details about the formulas and implementation are given in App. A.1 and App. A.2.

**Overview of the pipeline.** Fig. 1 summarizes the procedure. For a frozen pretrained LM $\mathcal{M}$, we run prefixes $\mathbf{x}_{1:T}$, select block index sets $\mathcal{S}$, and extract the corresponding hidden-state blocks $G_{\mathcal{S}}$. For each family of blocks, we then train a VQ-VAE so that its encoder maps each $G_{\mathcal{S}}$ to a discrete code $Z_{\mathcal{S}} \in [K]$, where $K$ is the codebook size (Step 1). Instead of fine-grained activations, we work with these high-level representation codes that serve as compact summary of complex internal computations in $G$. Importantly, we treat these codes not as semantically meaningful entities, but as representational summaries that allow us to find information relations. Therefore, the estimated mutual information depends not only on the underlying relationship between the hidden state themselves, but also on how they are encoded to high-level codes. Our use of VQ-VAE ensures that the learned representations are nontrivial since the decoder achieves meaningful reconstructions, and the cosine similarity penalty enforces diversity among codebook embeddings, which is validated by the reconstruction quality and codebook similarity results presented in App. A.1. In Step 2, we apply the trained encoder and codebook to the full dataset to obtain codes $Z_A$ and $Z_B$ for two compared block collections $G_A$ and $G_B$, count their co-occurrences. Then, we using the empirical joint and marginal distributions, in Step 3, we find $I(Z_A; Z_B)$ and its normalized variant $\text{nMI}(Z_A; Z_B)$, used as measures of how information is shared between different parts of the model's computation. Furthermore, we validate that our framework can find a closer relation to a known underlying distribution, in a representative experimental setting. Due to space limits, we refer the reader to App. A.4 for the details and results of the validation experiment. Experiments discussing probing-based method (Xu et al.) and a diagnostic illustrating why probing can have confounding effects are presented in App. D.2–D.5.

While our framework is general and could be applied to other applications in deep learning, in this work we use it to analyze how transformer models implement planning in practice, focusing on dimensions that are both theoretically meaningful and empirically informative.

## 3   Analysis of planning capability

We leverage our method to investigate two core planning-related aspects—(3.1) how much a model's prefix computation is forward-looking about future tokens, and (3.2) whether it preserves alternative correct continuations—and, as an application of the same framework, (3.3) how decision-relevant information is encoded across layers and earlier prefix blocks when producing next-token decisions. In our analyses, we use architecture-matched GPT-3 Small (Brown, 2020) models employing rotary position embeddings (Su et al., 2024). We report results from multiple seeds.

**Training objectives: next-token vs. multi-token.** In our analysis, we examine two variants of LM training objectives: $\mathcal{M}_{\text{NTP}}$ is trained with the standard next-token prediction (NTP) loss ($\mathcal{L}_{\text{NTP}}$), which optimizes conditional likelihood of the immediate next token, and $\mathcal{M}_{\text{MTP}}$ is trained with the multi-token prediction (MTP) loss ($\mathcal{L}_{\text{MTP}}$), which encourages models to align their hidden states with predictions across multiple forthcoming tokens. Following Gloeckle et al. (2024) for MTP implementation, we use shared transformer layers and separate heads for the next tokens during training. During inference, both models use standard autoregressive next-token generation. Intuitively, NTP prioritizes local token-by-token consistency, whereas MTP incentivizes models to form short-horizon plans and reduces strictly myopic behavior (Nagarajan et al., 2025; Bachmann & Nagarajan, 2024). Both losses are described in detail in App. B.1.

**Datasets.** Our experiments span three datasets chosen to cover distinct aspects of planning. *(i) A context-free grammar (CFG) dataset* (Allen-Zhu, 2024) emphasizes token-level rules and local

---

[2]In short, our VQ-VAE setting with the additional cosine loss combines the regularizing and contrastive approaches for representation learning, e.g., (Chen et al., 2020; Gao et al., 2021)

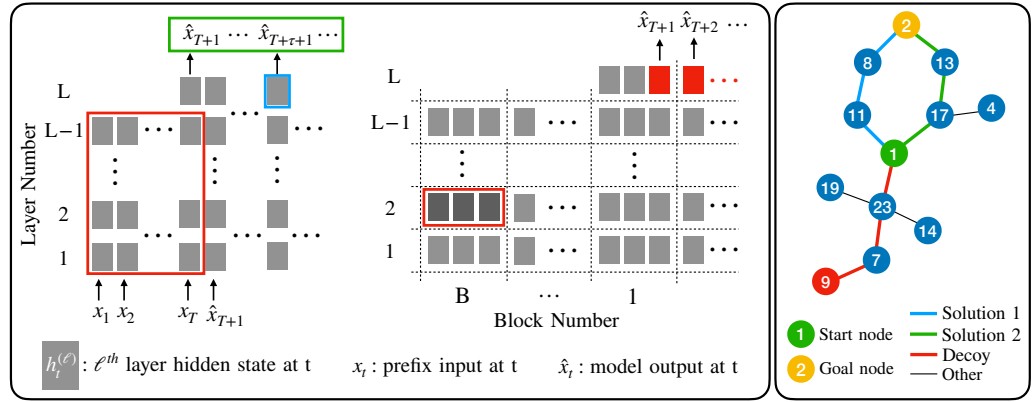

Figure 2: The visualizations of experimental settings. *Left:* Horizon of the plan and branches in the plan experiment (§ 3.1 & § 3.2). *Middle:* Information in the computational history experiment (§ 3.3). Colors in the *left* and *middle* panels denote the target variables. *Right:* An illustration of a simplified sample in PF task. Graphs and token numbers are randomly generated except for the start (1) and goal node (2). A sample prompt is "19 23 , 13 2 , 11 8 , 4 17 , 1 23 , 9 7 , 2 8 , 17 1 , 13 2 , 14 23 , 23 7 , 1 23 :" and correct responses are "1 11 8 2" or "1 17 13 2".

coherence. Tokens in CFG carry intrinsic meaning, much like words in natural language, and are governed by both local and global syntactic constraints. The model's task is to learn the grammar and predict valid next tokens that obey its rules. As a controlled substitute for natural language, CFG allows us to adjust vocabulary size and task difficulty, providing a clean environment for scientific study. *(ii) A path-finding (PF) task* (see Figure 2, right for illustration) requires models to generate valid paths between a fixed start and goal node in graphs with varying sizes and edge structures. The node vocabulary consists of 28 unique tokens, each representing a node identity. Unlike in CFG, these tokens do not carry intrinsic semantics and the meaning of a node token arises only through the set of edges it participates in. To construct language-model inputs, we linearize the graph by writing edges as pairs of adjacent node tokens separated by commas in a random order to prevent the model from exploiting positional shortcuts (e.g., "$u\ v,\ r\ t,\ \ldots$"). This edge-list representation ensures that the model must infer connectivity and graph structure rather than rely on token identity alone. Solving PF thus requires composing multiple reasoning steps over these relational inputs, analogous to chaining lemmas to form a proof, introducing a natural flavor of long-horizon planning. We construct two variants, PF-Short and PF-Long, which require finding paths of length 4 and 6 (including start and goal), respectively. *(iii) A natural language dataset*, OpenWebText (Gokaslan et al., 2019), reflects real-world distributional modeling. Here the model predicts continuations consistent with natural text. These datasets cover a spectrum from low-level symbolic structure to high-level naturalistic text. We refer the reader to App. B.2 for details.

## 3.1 HORIZON OF THE PLAN

In autoregressive generation, predictions for tokens beyond the immediate next one depend on both the prefix and the model's own generated outputs, which raises the question of how much computation the model allocates to tokens beyond the immediate prediction. We specifically ask:

*How much do LMs plan about future tokens before deciding on the immediate next token?*

To answer this question, we analyze two sets of activations. First, for a prefix of length $T$, we take the final-layer hidden states of the autoregressively generated tokens, $h_{T+\tau}^L$ for $\tau \geq 0$. As $h_{T+\tau}^L$ fully specifies the distribution over $\widehat{x}_{T+\tau+1}$, we treat it as the model's decision state and use its code $Z_{T+\tau}^L$ as a concise summary of the strategy employed for that token, obtained from a trained VQ-VAE. Second, we collect all prefix activations across layers:

$$H = \{ h_t^\ell \mid t = 1, \ldots, T; \ell = 1, \ldots, L-1 \} \in \mathbb{R}^{T \times (L-1) \times d},$$

where $H$ encodes the entirety of the model's prefix computation that has an effect on future token generations. Training a separate VQ-VAE over $H$, we get the LM's pre-output computation summary, and denote its high-level code as $Z_{1:T}^{1:L-1}$. $H$ and $h_{T+\tau}^L$ are respectively shown in red and blue colors in Figure 2 (left). We address our question by comparing how much the prefix

computation's summary $Z_{1:T}^{1:L-1}$ tells us about the model's decision state at the generated tokens. We measure the nMI between $Z_{1:T}^{1:L-1}$ and $Z_{T+\tau}^{L}$ for $\tau \geq 0$ as defined in Eq. 2 with normalization $\mathcal{I}_{\max} = \max\{\mathcal{I}(Z_{1:T}^{1:L-1}; Z_{T+n}^{L})\}_n$. If this ratio remains significantly above zero for large $\tau$, then the pre-output computation $Z_{1:T}^{1:L-1}$ carries information on the strategy to produce $\hat{x}_{T+\tau}$, indicating the model's initial prefix processing is not merely myopic but encodes forward-looking structure that persists into later generations. Conversely, if the ratio quickly decays to zero, then prefix computations primarily support only the immediate next prediction, with little evidence of long-horizon dependence between prefix computation and later decision states.

**Context-free grammar results.** We generate sentences of length $[16, 67]$ from CFG rules, and for each sample select a prefix length $T \in [8, 24]$ uniformly at random before, with the model generating the remaining continuation. For $\mathcal{M}_{\text{NTP}}$ and $\mathcal{M}_{\text{MTP}}$, Figure 3a reports the nMI between $Z_{1:T}^{1:L-1}$ and $Z_{T+\tau}^{L}$ across $\tau \geq 0$. Because CFG tokens have intrinsic meaning and are significantly governed by local syntactic constraints, it is expected that prefix computations correlate most strongly with the first few generated tokens. Indeed, $Z_{1:T}^{1:L-1}$ retains the highest information about the decision state for $\tau = 0$, and by $\tau = 10$ the nMI drops to roughly one-fifth of its initial value. This indicates that the pre-output computation encodes a short-horizon plan tied mainly to the next few tokens. We observe the same rapid decay of nMI when scaling the LM to $0.3$B parameters (App. D.1. $\mathcal{M}_{\text{MTP}}$ exhibits a similar pattern, with a slightly slower decay for $\tau < 10$, suggesting that the MTP loss does not substantially extend the horizon of pre-output computation in a model well-trained on CFG.

**Path finding results.** On PF-Short and PF-Long, we trained models on NTP and MTP losses. Because the start and end nodes are fixed tokens, the challenge lies in predicting the correct intermediate nodes ($\tau = 1, 2$ for PF-Short and $\tau = 1, 2, 3, 4$ for PF-Long) that connect them. We therefore focus our analysis on the intermediate positions and compute nMI as defined in Eq. 2. Figure 3b shows nMI across the generated path as well as the accuracy of correctly finding the whole path. For PF-Short, we see similar nMI trends across generated tokens for both $\mathcal{M}_{\text{MTP}}$ and $\mathcal{M}_{\text{NTP}}$, which both attain high accuracy. Interestingly, prefix computations encode more information about the *second* intermediate node than about the first. A likely explanation is that there is less uncertainty about the final token given the edge information and that the model spares more pre-output computation to find the last token. This resembles a strategy to work backwards from the goal, as a human solving this task might do. Similarly, for PF-Long, in Figure 3b, both $\mathcal{M}_{\text{NTP}}$ and $\mathcal{M}_{\text{MTP}}$ prefix computations encode a comparatively high amount of MI about the future tokens, unlike to the steady decay seen in CFG. This suggests that pre-output computations are not limited to the next token but also embed plans for upcoming positions, reflecting deliberate allocation of capacity toward future outputs. This finding is aligned with previous research on training LMs to exploit this reverse-solving approach, which has been shown to improve performance (Bachmann & Nagarajan, 2024). Taken together, these results indicate that LMs trained with both NTP and MTP loss can exhibit non-myopic behavior when trained on tasks that demand it. Lastly, in the PF-Long experiment, we observe that the nMI of $\mathcal{M}_{\text{MTP}}$ is slightly more uniform across $\tau$ than that of $\mathcal{M}_{\text{NTP}}$, which is a likely explanation for its higher accuracy by better predicting the earlier (harder) tokens along the generated path.

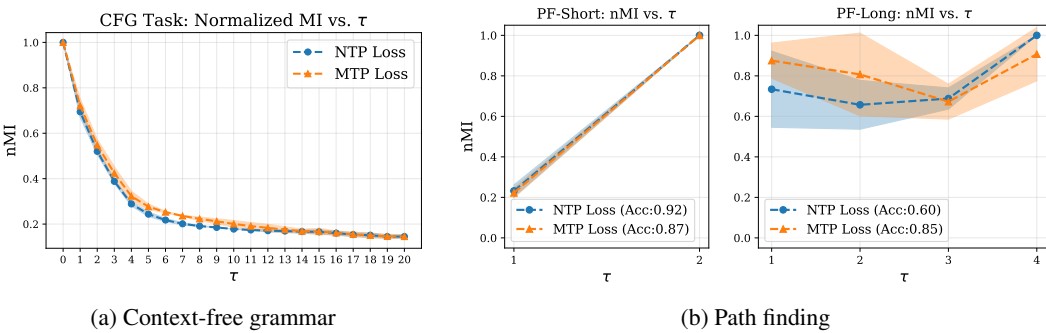

(a) Context-free grammar          (b) Path finding

Figure 3: nMI results between the prefix summary codes and the last hidden state codes of generated tokens for CFG (a) and PF (b) tasks. nMI decays fast in CFG, consistent with short-range dependence, while PF maintains or even increases nMI beyond $\tau = 1$, consistent with longer-horizon predictive dependence of prefix computation on later decision states.

Table 1: Mean $\pm$ std for $\mathcal{I}(Z_H; Z_{\text{alt}})/\mathcal{I}(Z_H; Z_{\text{decoy}})$ metric and accuracy values in branches in the plan experiment (§ 3.2). Models encode information about unchosen correct branches more strongly than unrelated decoys, indicating branch awareness in prefix computations.

| Model | PF-Short | | PF-Long | |
|---|---|---|---|---|
| | $\frac{\mathcal{I}(Z_H;Z_{\text{alt}})}{\mathcal{I}(Z_H;Z_{\text{decoy}})}$ | Accuracy | $\frac{\mathcal{I}(Z_H;Z_{\text{alt}})}{\mathcal{I}(Z_H;Z_{\text{decoy}})}$ | Accuracy |
| $\mathcal{M}_{\text{NTP}}$ | $7.60 \pm 0.78$ | $0.92 \pm 0.03$ | $1.45 \pm 0.01$ | $0.60 \pm 0.01$ |
| $\mathcal{M}_{\text{MTP}}$ | $6.29 \pm 0.17$ | $0.88 \pm 0.05$ | $1.82 \pm 0.27$ | $0.85 \pm 0.02$ |

Overall, the dependency between pre-output summaries $Z_{1:T}^{1:L-1}$ and decision states $Z_{T+\tau}^L$ confirms that the planning horizon is task-dependent. On CFG, nMI drops quickly with $\tau$ (short, local planning); on PF, it stays high and can peak beyond $\tau = 1$, allocating compute to later steps.

## 3.2 BRANCHES IN THE PLAN

In many tasks, there can be multiple correct continuations for a given query. For example, a math problem may admit different solution strategies, or a natural language prompt may allow several equally valid completions. This raises the question of whether LMs, when producing one correct answer, also implicitly encode information about other possible correct answers. Specifically, for the subset of samples the model solves correctly, we ask:

*Does the model consider alternative correct answers when generating the next token?*

To investigate this question, we use the PF datasets, where each sample is constructed to have two correct paths and one decoy path (see App. B.2.2). By design, the correct paths and the decoy path do not share any common nodes, ensuring no correlation among them. Similar to § 3.1, we summarize prefix computations $H$ into high-level codes $Z_{1:T}^{1:L-1}$ using a `VQ-VAE` encoder. In addition, we train a separate `VQ-VAE` to encode an entire path to find codes for the alternative correct path ($Z_{alt}$) and the decoy incorrect path ($Z_{decoy}$). $H$ and the generated path are visualized in Figure 2 (left) with red and green color, respectively. We then measure the MI between the prefix and alternative solution versus that between the prefix and decoy path through the ratio $\mathcal{I}(Z_{1:T}^{1:L-1}; Z_{alt})/\mathcal{I}(Z_{1:T}^{1:L-1}; Z_{decoy})$. A ratio above one means that the prefix computations encode more information about the alternative path than the decoy, indicating branch awareness in pre-output planning.

Table 1 reports this ratio together with path-finding accuracy. On PF-Short, both $\mathcal{M}_{\text{NTP}}$ and $\mathcal{M}_{\text{MTP}}$ yield ratios well above 1 and high accuracy, showing that the prefix retains information about unused correct branches on easier instances. On PF-Long, the ratios are smaller yet remain above 1, which indicates attenuated but persistent branch information as path length grows. $\mathcal{M}_{\text{MTP}}$ achieves both higher accuracy and a larger ratio than $\mathcal{M}_{\text{NTP}}$, suggesting that a model which solves the task more reliably also maintains richer branch-aware computation in its prefix states. Because the correct and decoy paths are disjoint, trivial overlap is ruled out as an explanation.

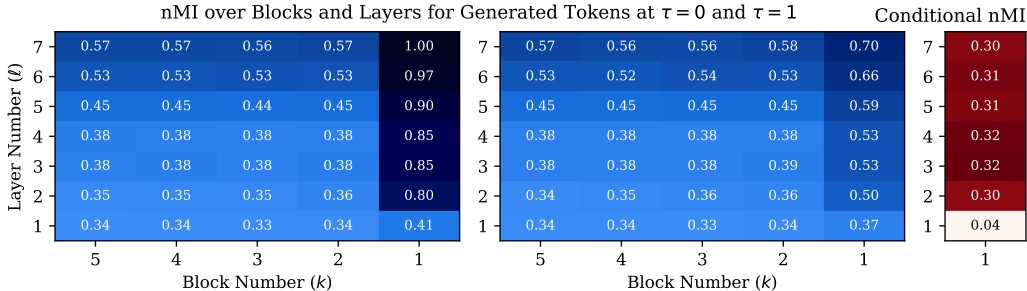

Figure 4: nMI across blocks and layers, and conditional nMI. *Left:* nMI between the hidden state block codes and the token decision state code at $\tau = 0$, $\text{nMI}(\mathbf{B}_k^\ell; Z_T^L)$. *Middle:* nMI between block codes and the last-layer decision code at $\tau = 1$, $\text{nMI}(\mathbf{B}_k^\ell; Z_{T+1}^L)$. In both heatmaps, nMI is higher for recent blocks (small $k$) and final layers (high $\ell$). *Right:* Conditional nMI for the 1st block, $\text{nMI}(Z_{T-15:T-1}^\ell; Z_T^L \mid Z_T^\ell)$, showing that most of the dependence between the 1st block and the generated token at $\tau = 0$ is attributable to the final prefix position $T$.

### 3.3 INFORMATION IN THE COMPUTATIONAL HISTORY

In an autoregressive LM, the final hidden state at the end of the prefix determines the next token, yet the computation producing it spans layers and earlier positions. This raises a natural question:

*Which earlier layers and prefix blocks remain informative about the next-token decision state?*

We view this primarily as a diagnostic. It does not tell us how to improve the model, but it quantifies how concentrated next-token decision information is in recent vs. earlier computation, which can help design choices that alter attention span or depth-wise context allocation. To answer the question, we use an 8-layer decoder-only Transformer trained on OpenWebText with NTP objective. We estimate the nMI between the codes of the model's decision state for the generated tokens and of blocks of prefix hidden state computations within the LM. Similar to § 3.1, we obtain a summary of the strategy employed for token $\widehat{x}_{T+\tau+1}$, denoted $Z^L_{T+\tau}$ with a VQ-VAE trained over the last layer of hidden states. We also partition each layer's prefix hidden states into contiguous, non-overlapping blocks of length 16 and train a VQ-VAE to acquire the code $\mathbf{B}^\ell_k$ where the block index $k \in \{1, \dots, 12\}$ denotes the $k^{\text{th}}$ block from the end of the prefix (i.e., $k = 1$ corresponds to the most recent 16 time steps before generation). In Figure 2 (middle), a sample block and a sample last hidden state are illustrated. We quantify the dependence between the codes of each computation block and the decision state by measuring nMI between $\mathbf{B}^\ell_k$ and $Z^L_{T+\tau}$ across all layers $\ell$ and blocks $k$, as defined in Eq. 2, with $\mathcal{I}_{\max} = \max\{\mathcal{I}(\mathbf{B}^\ell_k; Z^L_{T+n})\}_{\ell,k,n}$. Heatmaps in Figure 4 (left and middle) show the results for $\tau$= 0 and 1 (see App. B.5.1 for $\tau > 1$).

We observe that last layer computations retain the most information about the decision state of both immediate and future tokens which is aligned with the results of Pal et al. (2023) and consistent with the design choice of assigning longer attention spans to higher layers (Sukhbaatar et al., 2019; Beltagy et al., 2020). Furthermore, along the block axis, we observe a clear recency effect across all layers: the codes of the most recent blocks, i.e., final time steps of the prefix, computations ($Z_1$), exhibit the highest nMI (Eq. 2) with the decision state of the generated tokens, and the nMI decays over earlier blocks in the prefix. Although analysis along both axes indicates that the LM primarily relies on the most recent computations, we still observe appreciable nMI in lower layers (small $\ell$) and earlier blocks (large $k$). This suggests that LMs retain information from earlier parts of the prefix when generating new tokens, instead of relying only on the most recent computations.

We also define conditional normalized mutual information, $\text{nMI}(Z^\ell_{T-15:T-1}; Z^L_T \mid Z^\ell_T)$ (see App. B.5.2 for details), where $Z^\ell_{T-15:T-1}$ denotes the codes from layer $\ell$ spanning prefix positions from $T-15$ to $T-1$. This quantity, reported in Figure 4 (right), measures how much additional information about the decision state $Z^L_T$ is captured by earlier time steps in the 1st block, beyond what is already contained in the final prefix position $Z^\ell_T$. The resulting values are approximately 0.3, which is much lower than the unconditional nMI values of the 1st block observed in the left heatmap. This gap suggests that the majority of the dependency between the 1st block and the decision state arises from the final prefix token at position $T$. This finding further reinforces our earlier conclusion: *the LM primarily relies on the most recent computations when generating new tokens*.

## 4 CONCLUSION

Drawing on a VQ-VAE, we develop an information-theoretic pipeline that compresses hidden-state trajectories into discrete codes and uses them to compute MI across an LM's computation. This lens measures (i) how prefix computations inform future decision states, (ii) whether models retain information about alternative continuations, and (iii) how decision-relevant information is distributed across layers and prefix blocks. We evaluate models trained with standard NTP versus MTP losses on a synthetic CFG task, PF tasks, and natural text; we find no consistent differences between NTP- and MTP-trained models. The results are strongly task-dependent: in CFG, nMI between the prefix and the decision state decays quickly, consistent with largely myopic computation; in path-finding, the prefix retains substantial information about later steps and alternative correct paths, indicating stronger branch awareness. Across settings, we observe a recency effect, decision states depend most on late layers and recent blocks, while earlier activations remain measurably informative. Overall, LMs exhibit internal planning, but its extent and form vary with the task and training objective; our VQ-VAE-MI framework provides an automated way to study these behaviors. Future work could extend this analysis to reasoning models and test architectural changes that promote planning.

**Acknowledgements.** This work was partially supported by the US National Science Foundation under grants CCF 2045694, CNS-2112471, CPS-2111751, ONR N00014-23-1-2149 to GJ and US Department of Energy under grant DESC0025652 to CJW. In addition, the work was supported by Pennsylvania Infrastructure Technology Alliance, NSF Grants 2154171, CAREER Award 2339112, NSF Award 2512805, CMU CyLab Seed Funding to GQ. This work used PSC Bridges-2 GPU at Pittsburgh Supercomputing Center through allocation CIS250226 from the Advanced Cyberinfrastructure Coordination Ecosystem: Services & Support (ACCESS) program, which is supported by US National Science Foundation grants #2138259, #2138286, #2138307, #2137603, and #2138296.

**Reproducibility statement.** We describe our method in § 2, with further details provided in the relevant appendices. Our experimental analyses are presented in § 3, with additional details likewise included in the appendices. To ensure reproducibility, we provide the full code, tools to acquire the datasets, and instructions in the supplementary material. In addition, we include a README file that explains how to run each experiment separately.

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

## A  METHOD DETAILS

In this section, we explain the details of our method and implementation.

### A.1  VQ-VAE DESIGN AND IMPLEMENTATION

Let $h_t^\ell \in \mathbb{R}^d$ denote the hidden activation at token position $t$ after Transformer layer $\ell$. We view a computation block as a set of layer–token indices $S \subseteq \{1, \ldots, L\} \times \{1, \ldots, T\}$ and write the corresponding activations as

$$G_S = \{ h_t^\ell \mid (\ell, t) \in S \}.$$

The goal is to map the variable-shaped $G_S$ to a discrete code $z_S \in [K]$ with a vector-quantized variational autoencoder (VQ-VAE). These codes serve as coarse summaries that represent the computation block and make information-theoretic analysis feasible. We measure dependencies between computations using codes, for example $I(Z_A; Z_B)$ for two blocks $G_A$ and $G_B$, as in the main text § 2.

**Architecture overview.**  We train separate VQ-VAEs for target $S$ sets used in our analyses. Each VQ-VAE has an encoder $E$ that takes $G_S$ and outputs a latent vector $r_S \in \mathbb{R}^{d_e}$, a codebook $\{e_k\}_{k=1}^K \subset \mathbb{R}^{d_e}$, and a decoder $D$ that reconstructs $\widehat{G}_S$ from the selected codebook vector. Quantization uses nearest neighbors

$$k^\star = \arg\min_{k \in [K]} \|r_S - e_k\|^2, \qquad z_S \equiv k^\star, \qquad \tilde{r}_S \equiv e_{k^\star},$$

and the straight-through estimator for backpropagation is used for the gradient flow in training.

**Design rules by input structure.**  In our experiments, we select various $S$ depending on the analysis, as described in § 3. Even within the same experiment, $S$ can vary along the token axis because different samples may have different lengths. We design the encoder $E$ and decoder $D$ according to the structure of the hidden state blocks, which span the layer dimension, token index dimension, and hidden state vector dimension. Below we summarize the design specializations of $E$ and $D$ to explain how we handle challenges in the structure of the hidden state blocks.

- **Multiple layers at the same token index.** For each layer $\ell \in \mathcal{L}$, we first process the sequence $\{h_t^\ell\}$ with a transformer encoder. The resulting representations are then stacked across the layer axis. An MLP maps the stacked representation from dimension $|\mathcal{L}| \cdot d$ down to $d_e$ in the encoder. In the decoder, starting from $\tilde{r}_S$, we follow exact reverse operations, e.g., dimensionality expansion instead of dimensionality reduction with MLP.

- **Multiple token indices at the same layer.** We treat $\{h_t^\ell\}_{t \in \mathcal{T}}$ as a sequence and process it with a transformer encoder. To handle variable sequence lengths, the encoder either crops the sequence to a fixed window $T_{\text{enc}}$ or appends $m$ learned sentinel vectors and uses their outputs to form $r_S$. In the end, we concatenate representations across multiple time indices and apply an additional

MLP for further processing to get the encoder output. In the decoder, we first upsample the time dimension by mapping $\tilde{r}_S$ with an MLP to a maximum original sequence of length with $d$-dimensional vectors, then refine per-token reconstructions with a transformer encoder.

- **No token axis (single vector) or after reducing it as above.** When the input does not include the token dimension, or when it has been reduced as described above, we use only MLPs in both the encoder and decoder. The encoder outputs a single summary vector, and the decoder reconstructs the corresponding $G_S$ shape.

## A.2 TRAINING AND LOSSES OF VQ-VAE

**Training setup.** We train each VQ-VAE on hidden state blocks $G_S$ extracted from a fixed, pre-trained LM $\mathcal{M}$ over datasets used in our analyses. For each mini-batch, we sample sequences $\mathbf{x}_{1:T}$, feed $\mathcal{M}$ with inputs to get hidden activations $h_t^\ell$, assemble the target blocks $G_S$ according to the experiment. Then, we optimize encoder $E$, codebook $\{e_k\}_{k=1}^K$, and decoder $D$ while keeping $M$ frozen. This follows the standard VQ-VAE pipeline (Van Den Oord et al., 2017) adapted to variable-shape transformer hidden states.

**Objective.** Given an input block $G_S$ and encoder output $r_S = E(G_S) \in \mathbb{R}^{d_e}$, we quantize by nearest neighbor

$$k^\star = \arg\min_{k \in [K]} \|r_S - e_k\|_2^2, \qquad \tilde{r}_S = e_{k^\star},$$

and reconstruct $\widehat{G}_S = D(\tilde{r}_S)$ with straight-through gradients for the quantizer (Van Den Oord et al., 2017). The loss combines four parts

$$\mathcal{L} = \underbrace{\mathcal{L}_{\text{rec}}}_{\text{data fidelity}} + \underbrace{\lambda_q \, \mathcal{L}_{\text{vq}}}_{\text{quantization and commitment}} + \underbrace{\lambda_{\cos} \, \mathcal{L}_{\cos}}_{\text{codebook diversity}} + \underbrace{\lambda_{\text{ent}} \, \mathcal{L}_{\text{ent}}}_{\text{anti-collapse}} .$$

*Reconstruction* uses mean-squared error over the entries of $G_S$,

$$\mathcal{L}_{\text{rec}} = \|G_S - \widehat{G}_S\|_2^2.$$

*Quantization and commitment* follow Van Den Oord et al. (2017) with the straight-through estimator

$$\mathcal{L}_{\text{vq}} = \|\text{sg}[r_S] - e_{k^\star}\|_2^2 + \beta \, \|r_S - \text{sg}[e_{k^\star}]\|_2^2,$$

where $\text{sg}[\cdot]$ is stop-gradient and $\beta$ balances codebook learning and encoder commitment. *Codebook diversity* encourages distinct codes by discouraging cosine similarity among embeddings,

$$\mathcal{L}_{\cos} = \frac{1}{K(K-1)} \sum_{i \neq j} \left( \frac{\langle e_i, e_j \rangle}{\|e_i\|_2 \, \|e_j\|_2} \right)^2,$$

which spreads codebook vectors and yields more discriminative high-level codes, consistent with our use of codes as discussed in main text § 2. To avoid mode collapse of VQ-VAE (Zhao et al., 2024; Zhu et al., 2025), we further uses an *anti-collapse entropy* term which employs soft assignments computed from distances to the full codebook. Let

$$p_k = \frac{\exp(-\alpha \|r_S - e_k\|_2^2)}{\sum_{j=1}^K \exp(-\alpha \|r_S - e_j\|_2^2)},$$

then we minimize the negative entropy

$$\mathcal{L}_{\text{ent}} = -H(p) = \sum_{k=1}^K p_k \log p_k,$$

which promotes higher-entropy assignments during training and discourages mode collapse.

**Stabilizing codebook usage.** VQ-VAE training can suffer from codebook collapse (Zhao et al., 2024; Zhu et al., 2025). We adopt three complementary heuristics. (i) *Dead-code reset*: codes that have not been the argmin $k^\star$ for a fixed window are reinitialized to random encoder outputs from the current batch. (ii) *Codebook dropout*: with probability $0.1$ we temporarily mask a random subset of codebook vectors when forming nearest-neighbor distances in the forward pass of training, which encourages redundancy avoidance and better exploration. (iii) *Entropy regularization*: the $\mathcal{L}_{\text{ent}}$ term above explicitly pushes assignments away from degenerate peaks.

**Optimization details.** We use the Adam optimizer for training. Inputs at every token index and layer index are normalized before encoding.

**Evaluation metrics.** For quantitative checks of representation quality and nontrivial code usage, we report: (i) *nRMSE* between inputs and reconstructions. Let $H$ denote a vectorized view of $G_S$ and $\widehat{H}$ that of $\widehat{G}_S$. With $Z_H \in [K]$ the selected code and $\widehat{H} = D(e_{Z_H})$, define

$$\text{nRMSE} \;=\; \frac{\|H - \widehat{H}\|_2}{\|H\|_2}.$$

(ii) *Codebook geometry*: cosine similarity among codebook vectors $\{e_k\}_{k=1}^{K}$ and its distribution, which reflects diversity. (iii) *Usage distribution*: empirical frequencies of $Z_H$.

**Rationale for the objective.** The reconstruction term certifies that codes retain task-relevant information about $G_S$, the VQ term ties encoder outputs to discrete codes, the cosine penalty promotes discriminative summaries that support our information-theoretic analyses, and the entropy term plus stabilization heuristics sustain broad code usage throughout training.

### A.3 VQ-VAE TRAINING SETUP ACROSS EXPERIMENTS

Across all experiments, each `VQ-VAE` is trained once per dataset/task and representation type. This design ensures that the quantization mechanism remains fixed within each study, avoiding confounding effects that would arise from retraining `VQ-VAE` models at different mutual information estimates. In the planning horizon experiments, each underlying dataset is equipped with its own hierarchical pair of `VQ-VAE`s (`VQ-VAE`$_1$ and `VQ-VAE`$_2$) for quantizing prefix hidden-state blocks, along with its own last-layer `VQ-VAE`. Thus, while the architectural pattern is shared, the models themselves are trained separately for each task. In the branching experiments, we again train `VQ-VAE`$_1$ and `VQ-VAE`$_2$ for prefix hidden states, together with a dedicated path-sequence `VQ-VAE`. In the history experiments, we train one `VQ-VAE` for prefix blocks and one for last-layer OpenWeb-Text representations. Once trained, all `VQ-VAE` models are reused for every mutual-information estimation within their respective experimental setting.

With respect to **training budget**s, the `VQ-VAE` models require only moderate optimization steps. In the planning horizon experiments, `VQ-VAE`,1 and `VQ-VAE`,2 (used for prefix hidden states) are trained for 15,000 optimization steps each, corresponding to 0.96M sequences in total, while the last-layer `VQ-VAE` is trained for 10,000 steps (0.32M sequences). The path-finding version of this experiment uses the same budgets. These settings apply uniformly to both next-token and multi-token prediction evaluations. In the branching experiments, `VQ-VAE`,1 and `VQ-VAE`,2 (for prefix states) are trained for 20,000 (0.64M sequences) and 15,000 steps (0.48M sequences), respectively, and the path-sequence `VQ-VAE` is trained for 25,000 steps (0.80M sequences). Overall, this experiment uses between 0.32M and 0.96M sequences per `VQ-VAE` depending on the role. In the history experiments, we train two `VQ-VAE`s, one for prefix hidden-state blocks and one for last-layer hidden states, each for 10,000 optimization steps, corresponding to 0.64M sequences per model.

Choosing hyperparameters for our `VQ-VAE` models requires some initial effort, but not because the models are inherently sensitive. Instead, the difficulty arises from the fact that our `VQ-VAE` objective includes several additional components beyond the conventional formulation, such as cosine-push regularization, entropy-based loss, and codebook-reset mechanisms. These additions interact in nontrivial ways, and as a result the first working configuration for the first experiment takes time to establish simply because there is no pre-existing recipe for our variant of the objective. Once this initial configuration is found, however, the process becomes easier. We consistently find that once a reasonable setting is chosen for one model, closely related values work across all others with little to no retuning. The only exception is the codebook size. Starting from very small values, increasing the codebook size improves `VQ-VAE` performance, but only up to two possible limits. The first limit is model size. If the `VQ-VAE` is too small, increasing the codebook size provides little benefit, and making the model larger restores the usefulness of larger codebooks. The second limit is the inherent complexity of the task. After a certain point, increasing the codebook size and the model size no longer leads to meaningful improvements. For these reasons, our experiments use codebook sizes ranging from 64 (for quantizing a single token in a single-layer hidden state) up to 1024 (for

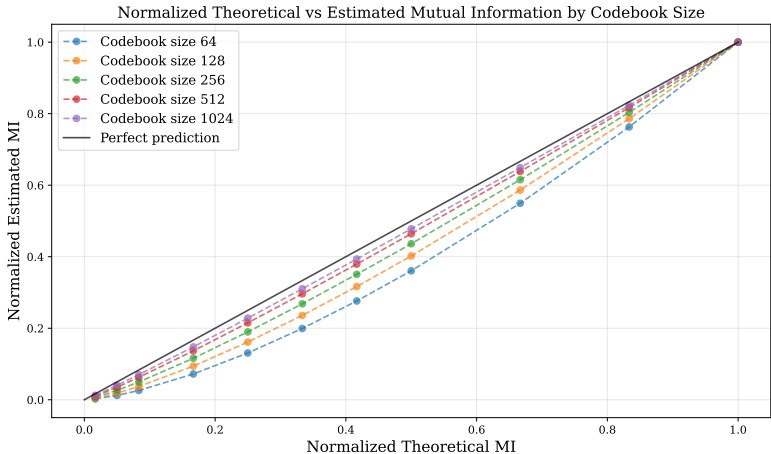

Figure 5: The comparison of our normalized latent mutual information estimations with different codebook sizes.

quantizing all prefix hidden states). For completeness and reproducibility, all hyperparameters can be accessed in the configurator files provided with the code.

### A.4 VALIDATION ON REPRESENTATIVE EXPERIMENTS

To validate our use of `VQ-VAE`-derived codes for estimating between computations, we conduct three experiments below with different difficulty levels.

#### A.4.1 VALIDATION EXPERIMENT WITH KNOWN MUTUAL INFORMATION

We define discrete variables $(A, B)$. Let $A$ be uniform on a domain $\mathcal{A}$ with $|\mathcal{A}| = 1024$, and let

$$\mathbb{P}(B = b \mid A = a) = \begin{cases} p_v & \text{if } b = a, \\ \frac{1-p_v}{1023} & \text{otherwise.} \end{cases}$$

The $\mathcal{I}(A; B)$ admits a closed-form expression that depends on $p_v$. We treat $(A, B)$ as the *ground-truth coarse variables* that our method aims to recover.

To mimic transformer computations, we associate each $a \in \mathcal{A}$ with a hidden-state tensor $G_A \in \mathbb{R}^{10 \times 11 \times 768}$ capturing all block outputs except the final layer of a trained GPT-3 small sized model given a random prefix of length 10. Also, we associate each possible value of $B$ with $G_B \in \mathbb{R}^{768}$, the final-layer hidden state at a single position. These tensors are *redundant surrogates* for $(A, B)$ in the sense that they only contain 10-bit information, i.e., $\log_2 1024$, despite their huge dimensions, which match the largest hidden-state structures used in our experimental analysis.

We then apply our pipeline: train `VQ-VAE` models on $G_A$ and $G_B$ to obtain discrete codes $Z_A$ and $Z_B$, which serve as coarse summaries of the corresponding computations. Finally, we compare $\mathcal{I}(Z_A; Z_B)$ to the ground-truth $\mathcal{I}(A; B)$ across values of $p_v$ and across codebook sizes $K$. Figure 5 reports both $\mathcal{I}(A; B)$ and $\mathcal{I}(Z_A; Z_B)$ normalized within each $K$. For every $K$, the codes preserve the ordering induced by the ground-truth , and $\mathcal{I}(Z_A; Z_B)$ approaches the ground truth as $K$ increases. We do not train to recover $(A, B)$, so $\mathcal{I}(A; B)$ is not a supervised target. Rather, $\mathcal{I}(A; B)$ is a *reference-process MI* that comes from the constructed latent variables defining the generative story behind $(G_A, G_B)$.

#### A.4.2 VALIDATION EXPERIMENT (HARDER SETTING WITH MULTIPLE SIMILAR PREFIXES FOR EACH LABEL)

We construct a more challenging variant of App. A.4.1. For each $a \in \mathcal{A}$ we now create a *set* of 16 surrogate hidden-state tensors in $\mathbb{R}^{10 \times 11 \times 768}$. Concretely, we take a random length-10 prefix and generate 15 additional prefixes by changing a single token at a random index, then run the trained

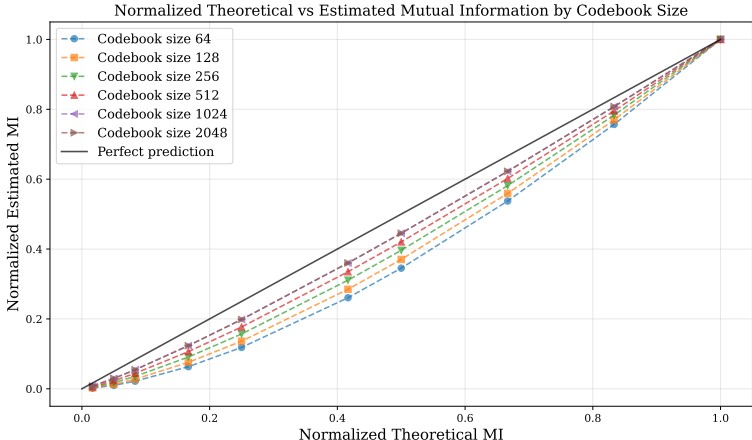

Figure 6: Harder validation experiment with similar prefixes for each label. Each label maps to 16 surrogates created by one-token edits. Normalized $\mathcal{I}(Z_A; Z_B)$ remains order-consistent with the reference-process MI across $K$, despite injected noise and within-class variability.

LM to collect the corresponding hidden state block outputs. Similarly, for each $b$ we produce 16 final-layer hidden states in $\mathbb{R}^{768}$ using the same single-token perturbation strategy. Therefore, this experiment consists of $2^{14}$ possible hidden state block $G_A$ and $2^{14}$ possible final-layer hidden state $G_B$. This setting reflects families of inputs that differ at one token yet retain essentially the same semantics, which is exactly the kind of fine detail our coarse summaries should compress away. To further increase difficulty, we add Gaussian noise during VQ-VAE training on hidden states. We then compute normalized mutual information between the learned coarse variables, treating the 16 surrogates for a fixed label as equiprobable when forming empirical distributions. The results in Figure 6 show that for every codebook size $K$, the codes preserve the ground-truth ordering across $p_v$, and $\mathcal{I}(Z_A; Z_B)$ tightens toward the reference as $K$ increases.

### A.4.3 VALIDATION EXPERIMENT (HARDEST SETTING WITH MULTIPLE UNRELATED PREFIXES)

We now remove within-class similarity entirely. For each $a \in \mathcal{A}$ we sample 16 *independent* length-10 token sequences uniformly at random and collect their $\mathbb{R}^{10 \times 11 \times 768}$ hidden-state tensors. For each $b$ we likewise sample 16 independent sequences and extract the corresponding $\mathbb{R}^{768}$ final-layer states. Thus, the 16 surrogates that share a label need not be close in vector space. This setting emulates many-to-one mappings from highly diverse inputs to the same high-level code. This setting resembles natural language settings with dissimilar texts having similar meanings. Again, this experiment consists of $2^{14}$ possible hidden state block $G_A$ and $2^{14}$ possible final-layer hidden state $G_B$.

We repeat the same pipeline and probability treatment as in App. A.4.2, estimating normalized MI between the learned coarse variables while assuming the 16 surrogates per label are equiprobable. Figure 7 reports the results. Although this regime poses 16 equally likely outcomes per input and eliminates structure exploitable by local similarity, our method still recovers normalized MI that preserves the correct ordering across $p_v$, with improved agreement at larger $K$. This regime would be quite challenging for a probing-type method to recover any meaningful information due to one-to-many mapping with totally unrelated targets.

## B DETAILS OF EXPERIMENTAL ANALYSES

We provide details of experimental settings and analyses here.

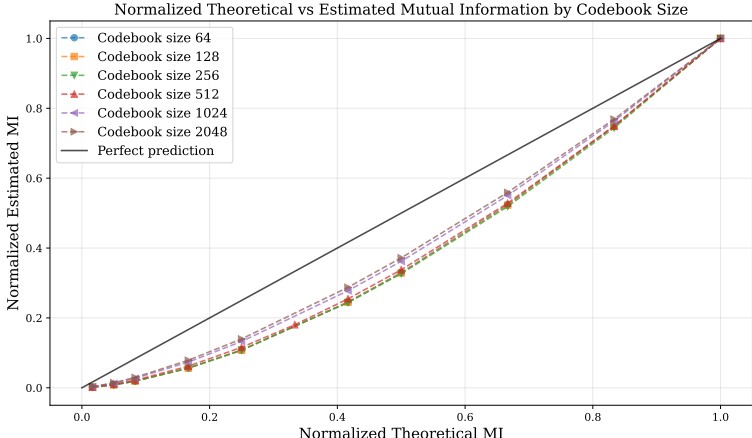

Figure 7: Hardest validation experiment with fully independent surrogates. Despite the lack of within-class proximity, normalized $\mathcal{I}(Z_A; Z_B)$ preserves the ordering induced by the reference-process MI, and improves as the codebook grows.

### B.1 LOSS FUNCTIONS

Let $\mathcal{V}$ be the vocabulary. A sequence is $\mathbf{x} = (x_1, \ldots, x_T)$ with prefixes $x_{\leq t}$. The model $\mathcal{M}$ has embedding $E \in \mathbb{R}^{|\mathcal{V}| \times d}$ and a shared unembedding $U \in \mathbb{R}^{|\mathcal{V}| \times d}$. The transformer has $L$ layers and width $d$. Hidden states are $h_t^\ell \in \mathbb{R}^d$ for $\ell = 1, \ldots, L$ and $t = 1, \ldots, T$; we write $h_t^L$ for the final-layer state. We use $\mathbb{P}$ for probabilities and $\log$ for natural logarithms.

**Next-token prediction (NTP).** With teacher forcing, the baseline objective maximizes the conditional likelihood of the immediate next token from the final-layer state $h_t^L$. We use a shared unembedding $U \in \mathbb{R}^{|\mathcal{V}| \times d}$:

$$\mathbb{P}_\theta\left(x_{t+1} \mid x_{\leq t}\right) = \text{softmax}(U\, h_t^L)_{x_{t+1}}, \qquad \mathcal{L}_{\text{NTP}}(\theta) = -\sum_{t=1}^{T-1} \log \mathbb{P}_\theta\left(x_{t+1} \mid x_{\leq t}\right).$$

**Multi-token prediction (MTP).** To align internal states with short-horizon futures, we equip the model with $\Gamma$ output heads Gloeckle et al. (2024). Head $\gamma \in \{1, \ldots, \Gamma\}$ is implemented as a transformer layer $f_h^{(\gamma)} \colon \mathbb{R}^d \to \mathbb{R}^d$ that takes $h_t^L$ as input and is followed by the same shared unembedding $U$:

$$\mathbb{P}_\theta(x_{t+\gamma} \mid x_{\leq t}) = \text{softmax}\left(U\, f_h^{(\gamma)}(h_t^L)\right)_{x_{t+\gamma}}.$$

The training loss sums the cross-entropy across positions and horizons:

$$\mathcal{L}_{\text{MTP}}(\theta) = -\sum_{t=1}^{T-\Gamma} \sum_{\gamma=1}^{\Gamma} \log \mathbb{P}_\theta(x_{t+\gamma} \mid x_{\leq t}).$$

All heads contribute equally. At test time decoding remains autoregressive; the auxiliary heads affect only the training signal and can optionally be used for speedups.

In our experiments, when we use MTP loss in CFG task, we used $\Gamma = 4$. As for, PF-Long and PF-Short tasks we used $\Gamma = 2$

### B.2 DATASET

Here we explain how we generated the dataset and how we used them in our LM training.

### B.2.1 CONTEXT-FREE GRAMMAR DATA GENERATION DETAILS

The specific formal details of our CFG are not central to the claims of this paper. We use it only as a controlled source of long, purely syntactic sequences. Readers who want a formal presentation and broader context can consult the CFG-based data generation reference Allen-Zhu (2024), on which our CFG data is based on. Here we provide the details of our specific CFG parallel to the naming convention in Allen-Zhu (2024).

We construct a layered grammar that expands strictly downward from a single start symbol to a set of terminals. The layers are organized as one start symbol, then four intermediate nonterminal layers of size three each, and finally a terminal inventory of sixty four symbols, summarized as the tuple (1, 4, 4, 4, 64). For every nonterminal we define a small menu of alternative productions; each nonterminal has between three and five alternatives, and each alternative rewrites the parent into two or three symbols drawn only from the next layer. This acyclic layout fixes the depth of any derivation and guarantees termination.

Once the grammar is fixed, each sequence is produced by starting at the start symbol and repeatedly expanding pending nonterminals until only terminals remain. Whenever a nonterminal has several alternatives, one is chosen at random. Because expansion always moves one layer down, sequences are long but tightly controlled in length. In the configuration used in the paper we generate four million sequences. The shortest contains 16 terminals and the longest 67 (the number of terminals corresponds to the sequence length), measured without begin or end markers. The resulting corpus contains 4M sequences (150M tokens).

### B.2.2 PATH-FINDING DATA GENERATION DETAILS

PF–Short is built to be a small but nontrivial path–finding task. In every instance the start node is labeled 1 and the goal node is labeled 2. We first lay down two *correct* trunks from the start to the goal. Each trunk has exactly two internal nodes, so a correct answer has the form 1, a, b, 2. We then lay down two *decoy* trunks that also begin at the start but end at fresh terminals that are not the goal. Decoys are the same length as the correct trunks, and the interiors of all trunks are disjoint. Up to this point the graph contains the start, the goal, two correct routes that meet only at those endpoints, and two look–alike alternatives that never reach the goal.

To make the local neighborhood around each trunk less revealing, we sprinkle branches on top of the trunks. We do this in one layer: for each eligible trunk node we independently add a small number of fresh children drawn from a Poisson distribution with mean one, and we cap the number of branch children per node at two. We never branch from the goal or from decoy terminals, and for the start node we reduce its branch allowance to account for the four trunk edges already attached to it, so its total degree remains bounded. If the instance would exceed the node budget of 28 distinct labels we discard it and resample.

Before writing the example to disk we randomly relabel every internal node using a permutation of the labels 3 through 28, while keeping 1 and 2 fixed. We also shuffle the edge list. The model never sees coordinates or orders, only an unordered set of undirected edges and an answer slot. A simplified typical prompt looks like

1 9, 7 2, 9 5, 1 3, 3 7, 5 2:

and either 1 9 5 2 or 1 3 7 2 is accepted as correct. In our experiments the PF–Short training split contains 16M samples generated with the recipe above. We also have a similar sized dataset to train corresponding vqvaes and a smaller validation data. Please also refer to Figure 2 (right) for a simplified network example.

PF–Long extends the horizon while keeping local clutter modest. We again fix the start to 1 and the goal to 2. We lay down two correct trunks, each with four internal nodes, so a correct answer has the form 1, a, b, c, d, 2. We then add one decoy trunk of the same length that begins at 1 but ends at a fresh non-goal terminal; interiors are disjoint. To add distractors we apply a two-layer branching pass: a light first layer around trunks followed by a very light second layer, with at most one branch child per parent. We never branch from the goal or from decoy terminals, and we reduce the start's branch allowance to respect the degree cap. If a sample would exceed the budget of 28 distinct node labels we discard and resample. Finally we randomly permute the internal labels within 3 through

28, keep 1 and 2 fixed, remap the paths, and shuffle the edge list. The training split contains 16M PF–Long examples generated with this recipe.

### B.2.3 NATURAL LANGUAGE (OPENWEBTEXT)

Finally, we include a large-scale natural language dataset (OpenWebText) to examine whether our information-theoretic analyses extend to unconstrained real-world data. Unlike the synthetic CFG or path-finding settings, this dataset reflects distributional structure from natural language and allows us to test whether planning signals identified in controlled tasks also emerge under standard pretraining conditions.

### B.2.4 HOW THE GENERATED DATA ARE USED TO TRAIN THE LANGUAGE MODELS

For the CFG and OpenWebText corpora we run conventional pretraining in separate setups. For CFG, we first generate many standalone sequences that satisfy the grammar. We then randomly shuffle these CFG sequences and concatenate them end to end to form a single text stream for that corpus. Training samples are taken by choosing a random starting offset in the CFG stream and slicing a fixed-length block equal to the model context window; the loss is computed on all tokens in the block except any optional boundary markers.

For OpenWebText, we apply the same pretraining recipe but within its own corpus only: documents are shuffled and concatenated to form an OpenWebText stream, blocks are sampled at random offsets, and next-token prediction is used with causal masking. There is no mixing or interleaving between CFG and OpenWebText at any stage.

For the path-finding data we use supervised finetuning rather than corpus streaming. Each example is a self-contained prompt and answer: the input is the unordered edge list written as comma-separated node pairs followed by a colon, and the target is any valid shortest path from the fixed start to the fixed goal. We concatenate prompt and answer for teacher forcing, mask the prompt so the loss is applied only to the answer span, and we do not concatenate different path-finding examples or split one across blocks. Under this scheme we use both PF–Short and PF–Long, whose correct answers have lengths four and six respectively when counting the start and goal.

### B.3 DETAILS OF PLANNING HORIZON EXPERIMENT (3.1)

**Details about VQ-VAE.** The VQ-VAE used to quantize the prefix representation $H \in \mathbb{R}^{T \times L \times d}$ employs a 256-entry codebook ($Z_H \in \{0, 1, \ldots, 255\}$), while each future final-layer state $h_\tau^L \in \mathbb{R}^d$ is quantized with a 64-entry codebook ($Z_{h_{T+\tau}^L} \in \{0, 1, \ldots, 63\}$).

For the CFG task, the dimension of $H$ is on average $16 \times 11 \times 768 = 135{,}168$, and the dimension of $h_\tau^L$ is 768. The mean nRMSE values of VQ-VAE encoding $H$ into summary codes is 0.47, and that of encoding $h_\tau^L$ is 0.21. Representative codebook similarity and usage plots from VQ-VAE training to encode $H$ for the CFG task are provided in Figure 8 and Figure 9. Also, those of VQ-VAE to encode $h_\tau^L$ for the CFG task are provided in Figure 10 and Figure 11

For the PF-Short task, the dimension of $H$ is on average $70 \times 11 \times 768 = 591{,}360$, and the dimension of $h_\tau^L$ is 768. The mean nRMSE values of VQ-VAE encoding $H$ into summary codes is 0.75, and that of encoding $h_\tau^L$ is 0.48.

For the PF-Long task, to reduce dimensionality, we subsample every three layers when encoding $H$ due to the limited model size, and we discard token indices corresponding to comma (,) tokens. The dimension of $H$ is on average $54 \times 4 \times 768 = 165{,}888$, and the dimension of $h_\tau^L$ is 768. The mean nRMSE values of VQ-VAE encoding $H$ into summary codes is 0.59, and that of encoding $h_\tau^L$ is 0.49.

*We emphasize that these are the results of huge quantization mappings, e.g., from $591{,}360$ real numbers to $256$ discrete labels.*

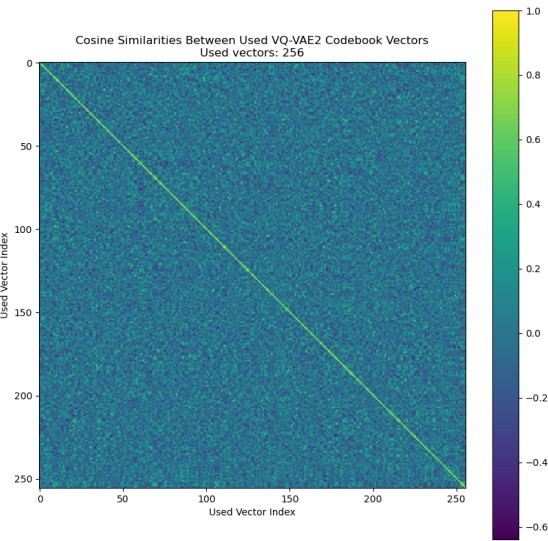

Figure 8: The codebook similarities for `VQ-VAE` encoding $H$ in CFG task.

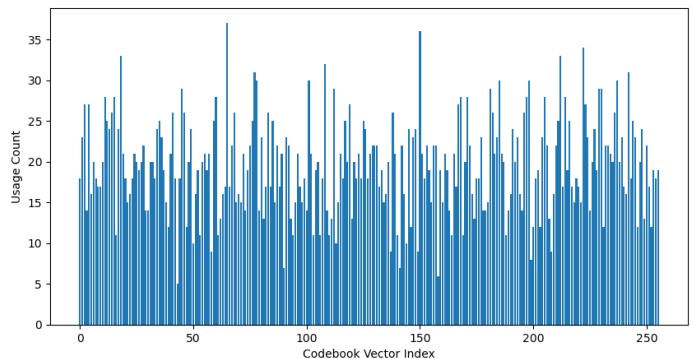

Figure 9: The codebook usage for `VQ-VAE` encoding $H$ in CFG task.

### B.4 DETAILS OF BRANCHES IN THE PLAN EXPERIMENT (3.2)

**Details about `VQ-VAE`.** For the `VQ-VAE` details encoding $H$ in PF-Short and PF-Long tasks, please, refer to the relevant part of App. B.3 since the same models are used.

**Encoding the paths.** The paths themselves does not have a vector representation and it is challenging use a unique class each possible path since in PF-Long, there can be $28^4$ paths. Although PF-Short is relatively easier, we reduce the number of possible codes to 512 by training `VQ-VAE` over a learned embedding vector for each of 28 tokens. Then we obtain the high-level codes of full path using the trained `VQ-VAE`. For PF-Long, we calculate the MI between the $H$'s code obtained from `VQ-VAE` and each intermediate token (using the token's index itself) along the response path. Then, we take the mean of these values over the nodes on the path to get a full-path MI result.

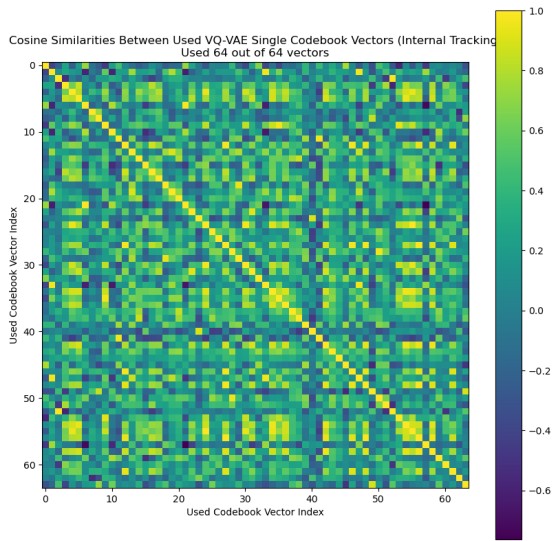

Figure 10: The codebook similarities for `VQ-VAE` encoding $H$ in CFG task.

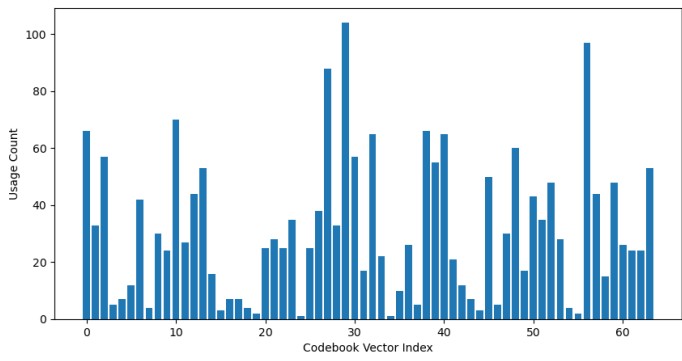

Figure 11: The codebook usage for `VQ-VAE` encoding $H$ in CFG task.

### B.5 DETAILS OF THE INFORMATION IN THE COMPUTATIONAL HISTORY EXPERIMENT

In this section we provide details of the information in the computational history experiment.

#### B.5.1 NMI RESULTS IN § B.4 FOR LONGER-HORIZON GENERATED TOKENS (HIGHER $\tau$)

We provide the results in Figures 12&13.

#### B.5.2 DETAILS OF THE CONDITIONAL NMI ESTIMATES

To estimate $\mathcal{I}(Z^{\ell}_{T-15:T-1}; Z^L_T \mid Z^{\ell}_T)$, we used the identity:

$$\mathcal{I}(Z^{\ell}_{T-15:T-1}; Z^L_T \mid Z^{\ell}_T) = \mathcal{I}(Z^{\ell}_{T-15:T}; Z^L_T) - \mathcal{I}(Z^{\ell}_{T-15:T-1}; Z^L_T), \quad (4)$$

which holds because

$$Z^{\ell}_{T-15:T} = \{Z^{\ell}_{T-15:T-1}, Z^{\ell}_T\}.$$

The first term of the right hand side of (4), is already estimated and shown in the Figure 4 ($k = 1$ block estimates). As for the second term, we find an approximate value as follows. First, because of

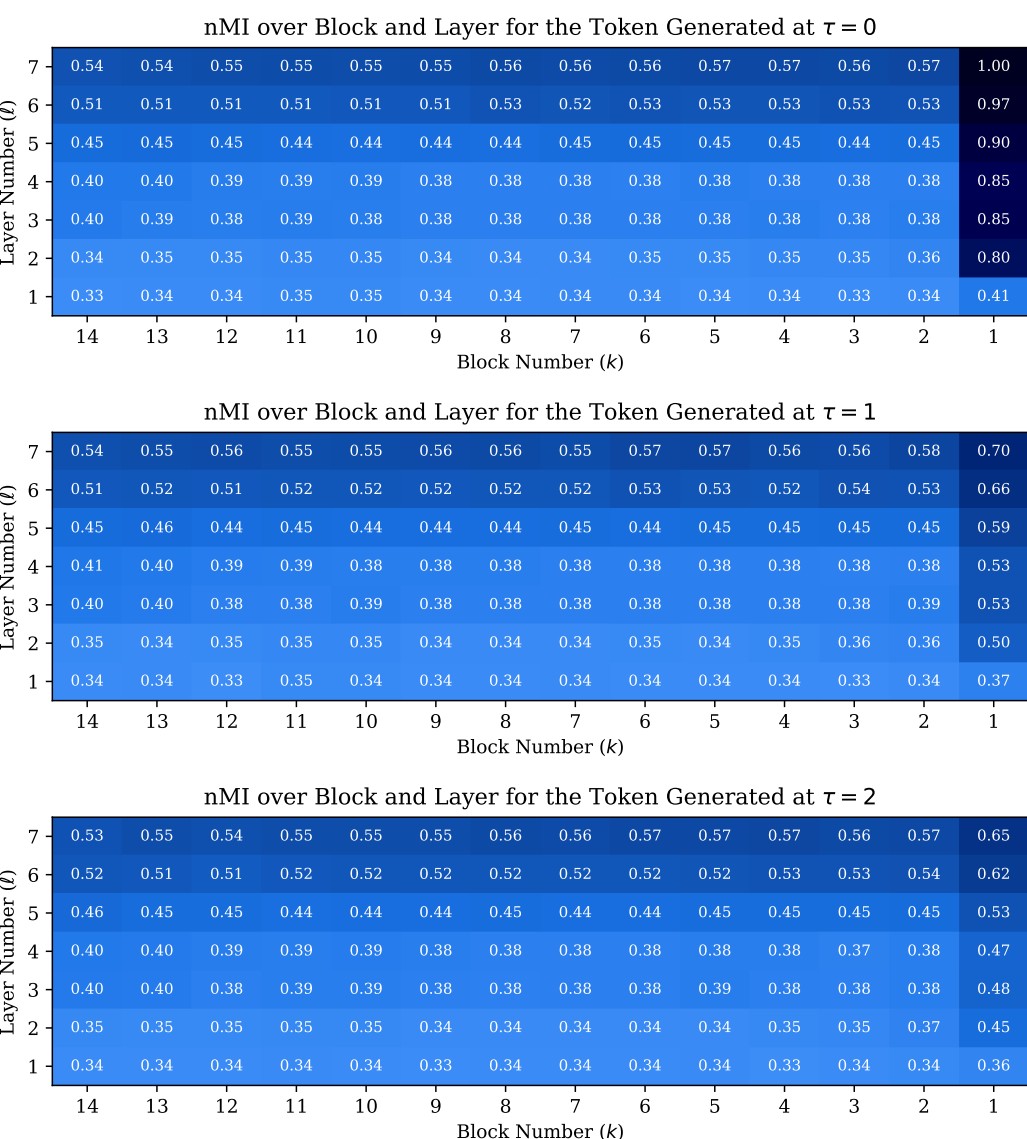

Figure 12: Heatmap of normalized mutual information over block and layer indices for $\tau = 0, 1, 2$. Please refer to the caption of 4.

the underlying task's symmetry (it is OpenWebtext, pretraining task), we have translational invariance, so

$$\mathcal{I}(Z^\ell_{T-15:T-1}; Z^L_T) = \mathcal{I}(Z^\ell_{T-14:T}; Z^L_{T+1}). \tag{5}$$

Then, we approximate the right-hand side of this equation by

$$\mathcal{I}(Z^\ell_{T-14:T}; Z^L_{T+1}) \approx \mathcal{I}(Z^\ell_{T-15:T}; Z^L_{T+1}),$$

which relies on the observation that in natural language, the incremental information contributed by extending the context diminishes: as sentences become longer, the mutual information between past tokens and future ones grows only logarithmically.

### B.5.3 DETAILS ABOUT BLOCK VQ-VAE.

The VQ-VAE used to quantize $\ell^{th}$ layer $k^{th}$ block representation $h^\ell_{T-16B:T+16-16B} \in \mathbb{R}^{16 \times d}$ employs a 1024-entry codebook ($\mathbf{B}^\ell_k \in \{0, 1, \ldots, 1023\}$), while each future final-layer state $h^L_\tau \in \mathbb{R}^d$

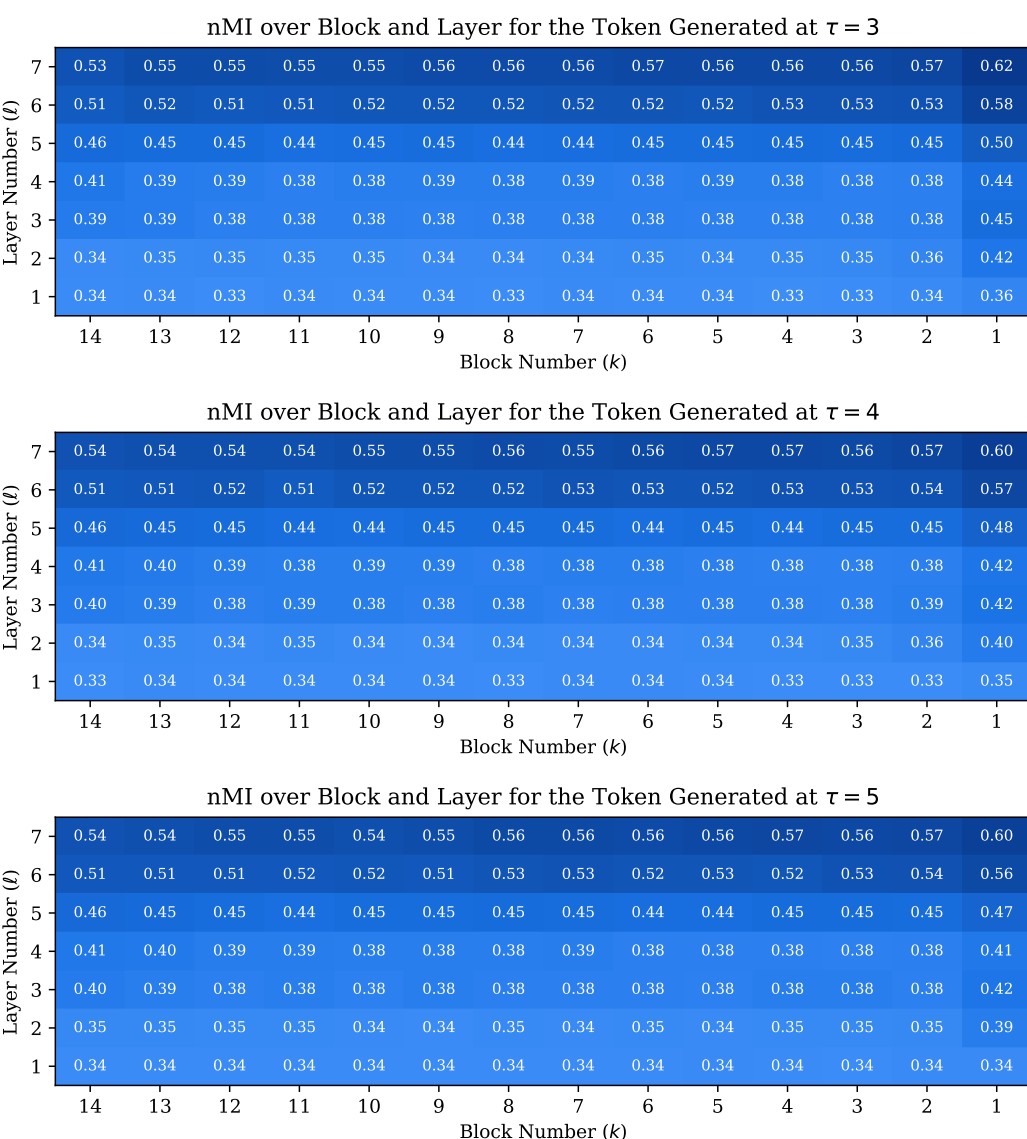

Figure 13: Heatmap of normalized mutual information over block and layer indices for $\tau = 3, 4, 5$. Please refer to the caption of 4.

is quantized with a 64-entry codebook ($Z_{h_{T+\tau}^L} \in \{0, 1, \dots, 63\}$). Recall this experiment is on NLP (OpenWebText pretraining). The dimension of $h_{T-16B:T+16-16B}^\ell$ is on average $16 \times 512 = 7680$, and the dimension of $h_\tau^L$ is 512. The overall VQ-VAE for the blocks has 4M parameters in total. Representative codebook similarity and usage plots from VQ-VAE training to encode $h_{T-16B:T+16-16B}^\ell$ for the (OpenWebText) task are provided in Figure 14 and Figure 15. For additional details on VQ-VAE trainings on OpenWebText (NLP pretraining) data please refer to App. B.6.

## B.6 ON THE SCALABILITY OF OUR PIPELINE

Recall that to obtain the code $Z_{T+\tau}^L$, we use an encoder whose input is $h_{T+\tau}^L$. To study this, we trained VQ-VAEs of increasing sizes on OpenWebText, with a fixed codebook size of 64 vectors. OpenWebText is widely used for NLP pretraining, but the data lacks a strong coherent structure across samples. As a result, it is natural, and in fact expected, that reconstruction errors remain relatively high.

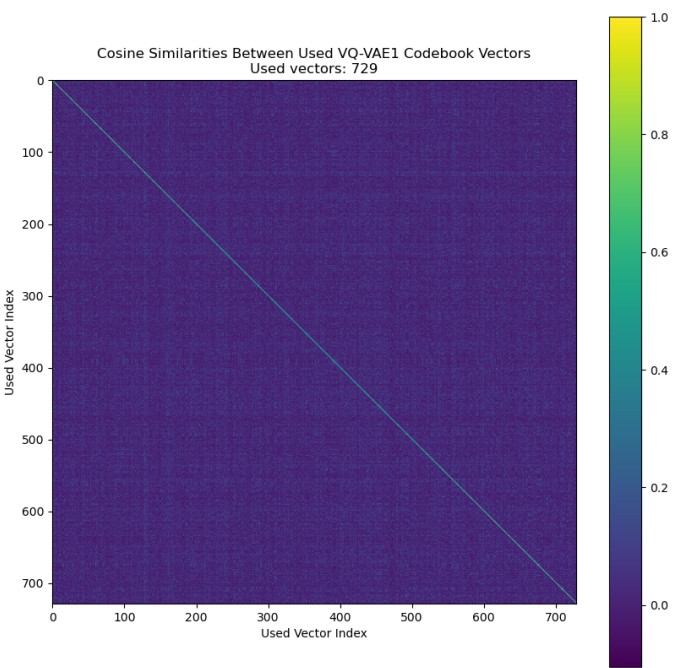

Figure 14: The codebook similarities for $\mathtt{VQ-VAE}$ encoding $h^\ell_{T-16B:T+16-16B}$ in OpenWebText task.

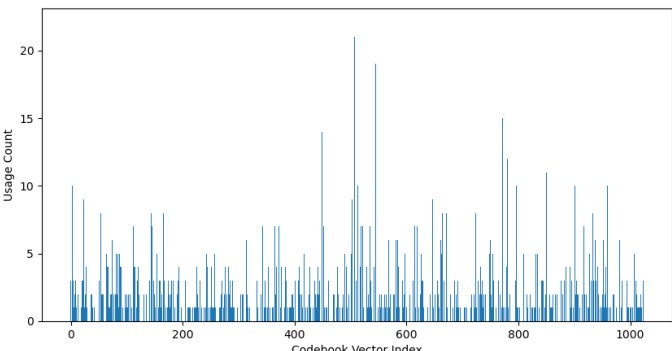

Figure 15: The codebook usage for $\mathtt{VQ-VAE}$ encoding $h^\ell_{T-16B:T+16-16B}$ in OpenWebText task.

Figure 16 reports the normalized nRMSE as a function of model size. This hints that scaling the encoder can improve representation quality despite the inherent noisiness of the dataset.

## C  RELATED WORK IN DETAIL

We provide a longer discussion of the Related Work in the main text § 1.

**Language model planning.**  In order to improve LMs' reasoning and planning abilities, researchers have developed scaffolding and augmentation techniques, including Chain-of-Thought prompting (Wei et al., 2022), planning tokens (Wang et al., 2024c; Sel et al., 2025), and structured inference methods such as Tree-of-Thoughts and Graph-of-Thoughts (Yao et al., 2023; Besta et al., 2024). In parallel, hybrid systems that integrate LMs with symbolic planners or external tools have achieved state-of-the-art performance in embodied and tool-use domains (Zhao et al., 2023; Wang et al., 2024a; Shen et al., 2023). Despite these advances, recent studies highlight that significant challenges remain, and current approaches to LM planning still fall short of fully addressing complex reasoning and decision-making tasks. For example, Lin et al. (2025) show that models can

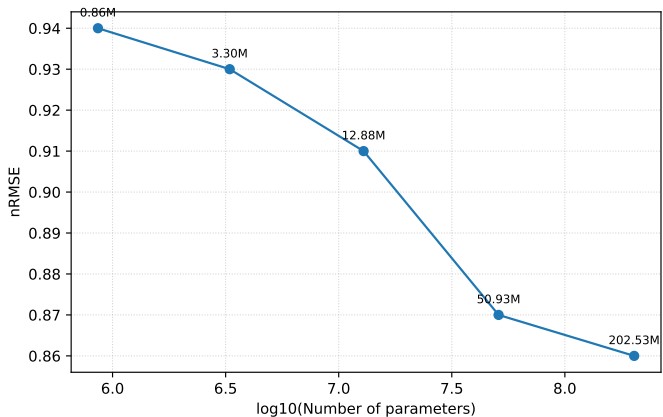

Figure 16: The nRMSE vs `VQ-VAE` encoder parameter count plot.

produce conflicting answers under logically related prompts despite local plausibility, which shows lack of long horizon planning, although short-horizon planning works fine; Ahn & Yin (2025) reveal discrepancies when asking a model what is correct vs what is incorrect; and Saxena et al. (2024) demonstrate that even repeated queries yield inconsistent outputs. Beyond inconsistency, some studies find that out-of-the-box models struggle with even simple planning tasks (Momennejad et al., 2023). Complementing these empirical findings, theoretical analyses suggest that autoregressive models may face fundamental limits in their planning ability (Wang et al., 2024b). Taken together, these findings underscore that understanding whether and how planning arises in LMs is not only an open empirical challenge, but also central to both their interpretability and the principled design of future model architectures.

**Behavioral and Mechanistic Interpretability.** Some interpretability (explainability) methods consider the LM as black box, and design tasks or benchmarks to gauge reasoning, robustness, or generalization abilities (Srivastava et al., 2023; Liang et al., 2022). In contrast to them, mechanistic interpretability seeks to reverse-engineer transformer computations into human-understandable parts, treating the residual stream as the main information pathway and attention heads as separate components that pass information along (Elhage et al., 2021). Using this approach, researchers have explained key phenomena in LMs: in-context learning can arise from a characteristic two-head circuit that appears during a training "phase change," with causal support from perturbation studies (Olsson, 2022); many neurons are polysemantic, but sparse autoencoders (SAEs) can replace them with more interpretable, monosemantic features that enable finer-grained causal analysis (Cunningham et al., 2023; Bricken et al., 2023); models implicitly encode structural linguistic properties such as syntax (Hewitt & Liang, 2019). Building on earlier approaches, transcoders, approximate dense MLPs with wider sparse layers, separating circuits into input-invariant and input-dependent components, and matching or surpassing SAEs in sparsity, faithfulness, and interpretability (Dunefsky et al., 2024). Frontier-scale case studies use cross-layer transcoders to trace multi-step reasoning and other behaviors, illustrating how these tools can audit mechanisms, not just features, in modern LLMs (Lindsey et al., 2025). Overall, mechanistic interpretability offers concrete tools for uncovering how LLMs compute, though these methods are still developing.

**Mathematical Perspectives.** Beyond empirical tools, mathematically grounded perspectives also illuminate transformer/LLM interpretability: automata-theoretic analyses show self-attention can implement finite-state algorithms (Liu et al., 2023); optimization views prove in-context learning corresponds to (preconditioned) gradient steps (Ahn et al., 2023) and transformers are mutual interaction learners (Ustaomeroglu & Qu, 2025); mean-field theory gives global convergence guarantees at scale (Gao et al., 2024).

**Probing.** A different line of interpretability work views LLM hidden states as structured representations that can be "probed", probing refers to training lightweight models, often linear classifiers, to

read out specific information from hidden states in order to test what the model represents internally. Using probing, researchers have shown that transformer hidden states encode structured belief-state and world-model–like information (Shai et al., 2024; Gurnee & Tegmark, 2024; Hazineh et al., 2023). Other works demonstrate that a single hidden state can carry information about multiple future tokens (Pal et al., 2023) and that probing can reveal the underlying algorithms LLMs use to solve tasks (Allen-Zhu, 2024).

However, standard accuracy-based probing has drawn a critique for combining what the probe can learn with and what the representation actually encodes making the results sensitive to probe capacity, data size, and hyperparameters Hewitt & Liang (2019); Pimentel et al. (2020). Further, high probe scores often come from exploiting superficial linear context cues rather than genuine structural knowledge (Kunz & Kuhlmann, 2020). Probing can even reveal features that a model does not use for its task (Ravichander et al., 2021; Kumar et al., 2022). Consequently, some critiques motivate adopting information-theoretic lenses (Voita & Titov, 2020; Diego-Simón et al., 2025). Several approaches illustrate this perspective. The original Information Bottleneck framework (Tishby & Zaslavsky, 2015; Shwartz-Ziv & Tishby, 2017) casts learning as a trade-off between compressing input representations and preserving predictive information about outputs. Complementary work by Voita et al. (2019) examined how information flows across transformer layers under different training objectives, revealing that tasks like language modeling, masked language modeling and machine translation induce distinct patterns of information loss and reconstruction. More recently, Skean et al. (2025) demonstrated that intermediate layers of LLMs often yield stronger representations than the final layer, using unified metrics based on entropy.

## D    EXTRA EXPERIMENTS WITH LARGER MODELS AND COMPARISON WITH PROBING

In this section we (i) scale experimented language model and repeat the *Horizon of the Plan* analysis, (ii) add probe-based baselines, and (iii) explain why probe performance can be confounded in our setting.

### D.1    SCALING TO A LARGER MODEL

We scale the GPT-3–based decoder-only architecture with Rotary Positional Encoding (Su et al., 2024) to 24 layers and $d_{\text{model}} = 1024$ ($> 0.3$B parameters), more than twice the size used in the main experiments. We rerun the *Horizon of the Plan* experiment on the CFG dataset from Section 3.1. The 0.3B model attains near-perfect sequence-completion accuracy.[3] Our method scales smoothly with larger language models.

Figure 17 reports the normalized mutual information between the prefix summary codes and the last-layer hidden-state codes at generated positions $t + \tau$. The nMI curve decays rapidly as $\tau$ increases, matching the trend in smaller models. On CFG, this is consistent with myopic behavior rather than long-horizon planning, which aligns with the task's syntactic structure.

### D.2    PROBING EXPERIMENTS

**Setup.** Probing fits a supervised predictor from internal states to a target and uses generalization performance as a proxy for whether the information is linearly or simply recoverable. We evaluate probes in the same setting as §D.1.

Let $L$ be the number of layers. For a prefix ending at time $t$, denote by

$$H_{\text{pref}} \triangleq \left\{ h_{\ell,i} \in \mathbb{R}^d \ : \ \ell = 1{:}L, \ i \leq t \right\}$$

the block of hidden states across all layers and prefix positions. For a future offset $\tau \geq 1$, denote the generated token by $\widehat{x}_{t+\tau}$ and the last-layer hidden state by $h_{t+\tau}^L \in \mathbb{R}^d$. We train separate probes for each $\tau$. Since the prefix length can vary across samples, we use zero padding for shorter samples.

---

[3]Sequence-completion accuracy is the fraction of prefixes for which the model completes a CFG-valid continuation.

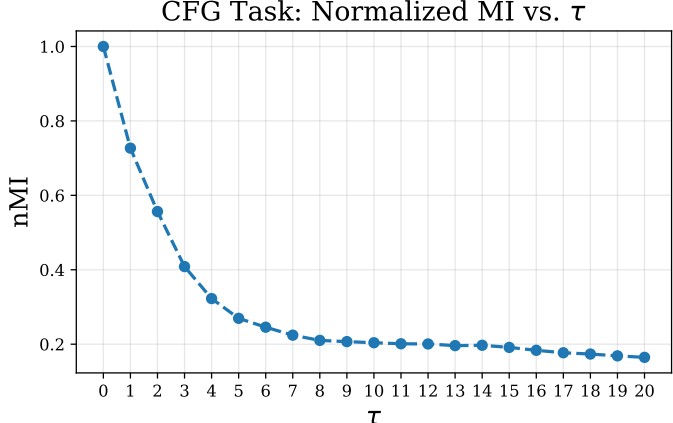

Figure 17: Normalized MI between prefix summary codes and the codes of the last-layer hidden state at future position $t + \tau$ on CFG for the $0.3$B LM. The rapid decay in nMI with $\tau$ replicates the main-text trend at larger scale.

**Token prediction probe.** A two-layer MLP $\phi_\tau$ maps $H_{\text{pref}}$ to a distribution over the 32 CFG tokens. The loss is cross-entropy,

$$\mathcal{L}_\tau^{\text{tok}} \;=\; \mathbb{E}\big[-\log p_{\phi_\tau}\big(\widehat{x}_{t+\tau} \mid H_{\text{pref}}\big)\big],$$

and we report accuracy.

**Hidden-state regression probe.** A two-layer MLP $\psi_\tau$ predicts the future last-layer hidden state,

$$\mathcal{L}_\tau^{\text{reg}} \;=\; \mathbb{E}\big[\|\psi_\tau(H_{\text{pref}}) - h_{t+\tau}^L\|_2^2\big].$$

We report normalized $\text{MSE}_\tau$, measured by dividing the MSE values by the highest MSE in that setting.

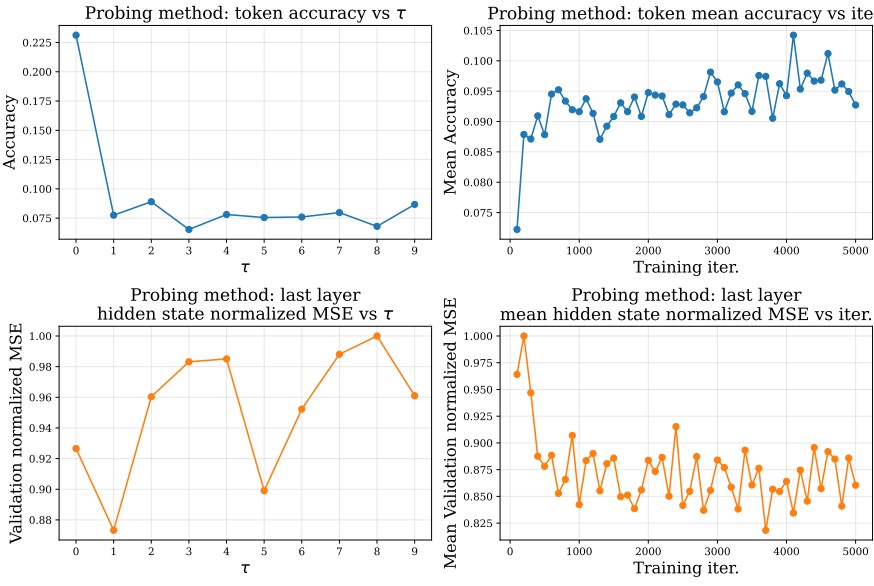

Figure 18: Probe baselines on CFG. *Top left:* token accuracy vs. future offset $\tau$. *Top right:* mean token accuracy vs. training iteration. *Bottom left:* hidden-state regression normalized $\text{MSE}_\tau$ vs. $\tau$. *Bottom right:* normalized mean MSE vs. training iteration.

**Findings.** Token probes are decisively above random sampling only for $\tau = 0$; beyond the immediate next token the accuracy does not change much and not yield a clear picture for interpretation. Hidden-state regression probes close normalized $\text{MSE}_\tau$ across $\tau$. Even with nonlinear two-layer MLPs and per-$\tau$ training, the results are inconclusive about the structure of long-horizon computation and planning in the language model. This reflects a practical limitation of probe-style supervision when the input $H_{\text{pref}}$ and the target $h_{t+\tau}^L$ are both high dimensional.

**Relation to linear or higher-capacity probes.** A linear probe is a special case of the MLP used here, so the nonlinear results upper bound what a linear probe can extract under the same inputs and targets. If high-capacity probes are used, supervised training can memorize dataset idiosyncrasies and fit mappings from $H_{\text{pref}}$ to the target that the LM itself does not implement, yielding optimistic scores without evidencing a mechanism in the model. These effects blur the link between probe performance and information flow.

We include probes as a *diagnostic comparison*, not as an estimator or validation of our MI measure. Probes answer a different question—how well a chosen supervised predictor can recover a hand-specified target from $H_{\text{pref}}$—and their performance can be sensitive to predictor capacity and target complexity. In our setting, probing does not yield a stable or interpretable signal about long-horizon structure across $\tau$, whereas our MI analysis measures statistical dependence between learned compressed summaries. See Appendix D.3 for an illustration of why probe scores can be confounded.

## D.3   WHY PROBING CAN BE CONFOUNDED AND A SIMPLE ILLUSTRATION

**Target-variance sensitivity.** Prior work (Voita & Titov, 2020; Diego-Simón et al., 2025) has noted that probe performance can be dominated by the marginal complexity of the target rather than by information shared with the source.

Let $X \sim \text{Unif}[0, 10]$ so $H(X) = \log_2 10 \approx 3.32$ bits. Define

$$Y_1 = 0, \qquad Y_2 = \begin{cases} X & \text{with prob. } 0.5, \\ 0 & \text{with prob. } 0.5. \end{cases}$$

Then $I(X; Y_1) = 0$ while $I(X; Y_2) = \frac{1}{2}H(X) \approx 1.66$ bits. A probe achieves zero error on $Y_1$ yet struggles on $Y_2$, which would wrongly suggest that $X$ contains more information about $Y_1$ than $Y_2$. The discrepancy arises because probe loss is governed by the target's marginal variability, not by the actual information shared with $X$. Furthermore, as discussed in the previous section, higher capacity probes can blur the line between dataset idiosyncrasies and actual language model mechanisms.

**Why our method avoids this pitfall.** Our method trains two VQ-VAE encoders $E_1$ and $E_2$ on prefix and future hidden states using only reconstruction losses, with no labels or task supervision. We then estimate mutual information between $\mathcal{I}\left(E_1(h_1); E_2(h_2)\right)$ and interpret it *comparatively* in all tasks. Our conclusions are comparative within a fixed experimental setup: we hold the encoders, codebooks, and MI estimator fixed and report normalized MI between the same representation types while varying only the variables of interest. For example, when comparing the MI between the summary of pre-output computations and the summaries of the first vs. second output token, every pipeline component is identical, so any MI shift must arise from the underlying interaction among the selected variables. Finally, by obtaining unsupervised, high-level representations, we reduce the effect of any blur induced by target-specific idiosyncrasies in the raw hidden states. Our method coarsens the space and the interaction among the underlying variables is not probe-induced shortcuts.

## D.4   PREDICTIVE $\nu$-INFORMATION CAN BE ARBITRARILY DISTORTED BY RESCALING

Xu et al. define predictive $\nu$-information $I_\nu(X \to Y)$ from a squared-loss prediction problem over a Gaussian linear family. In the setting of their Proposition 1.5, this reduces to

$$I_\nu(X \to Y) \;=\; \text{tr}\big(\text{cov}(Y)\big) R^2,$$

where $R^2$ is the coefficient of determination of the optimal linear regression of $Y$ on $X$.[4] Unlike Shannon mutual information, this quantity is sensitive to the marginal scale of $Y$ and can be changed arbitrarily by rescaling the target, even when the underlying information content is fixed.

To illustrate, let $X \sim \mathcal{N}(0,1)$ and $\epsilon \sim \mathcal{N}(0,1)$ independent. Define

$$Y_1 = X + \epsilon, \qquad Y_2 = a\,Y_1 = a(X + \epsilon),$$

for some scalar $a \neq 0$. Since $Y_2$ is an invertible linear transformation of $Y_1$, Shannon mutual information is invariant:

$$I(X; Y_1) = I(X; Y_2).$$

Now consider $I_\nu$ in the Gaussian linear setting of Xu et al. We have

$$\mathrm{Var}(Y_1) = \mathrm{Var}(X) + \mathrm{Var}(\epsilon) = 2, \qquad \mathrm{Var}(Y_2) = a^2 \mathrm{Var}(Y_1) = 2a^2.$$

The optimal linear predictor of $Y_k$ from $X$ is proportional to $X$ for both $k = 1, 2$, and the squared correlation is

$$R_1^2 = R_2^2 = \frac{\mathrm{Cov}(X, Y_1)^2}{\mathrm{Var}(X)\mathrm{Var}(Y_1)} = \frac{1^2}{1 \cdot 2} = \tfrac{1}{2},$$

since scaling $Y$ does not change correlation. Therefore,

$$I_\nu(X \to Y_1) = \mathrm{tr}(\mathrm{Cov}(Y_1))\,R_1^2 = 2 \cdot \tfrac{1}{2} = 1,$$

while

$$I_\nu(X \to Y_2) = \mathrm{tr}(\mathrm{Cov}(Y_2))\,R_2^2 = 2a^2 \cdot \tfrac{1}{2} = a^2.$$

By choosing $|a|$ arbitrarily large or small, we can make $I_\nu(X \to Y_2)$ arbitrarily larger or smaller than $I_\nu(X \to Y_1)$, despite the fact that $Y_1$ and $Y_2$ contain exactly the same Shannon information about $X$. This demonstrates that predictive $\nu$-information, in this common probe setting, fails to satisfy a basic information-theoretic property: invariance under invertible transformations of the target (equivalently, it does not obey the data-processing inequality with respect to deterministic rescalings of $Y$).

**Why our method avoids this scaling pathology.** In contrast, our VQ-VAE approach estimates Shannon mutual information between learned discrete codes $\mathcal{I}\left(E_1(h_1); E_2(h_2)\right)$. Because mutual information is invariant under invertible transformations of each argument and satisfies the data-processing inequality, rescaling or reparametrizing the underlying continuous hidden states before encoding cannot arbitrarily inflate or deflate the measured dependence. Once the encoders and codebooks are fixed, changes in our estimated MI across conditions reflect differences in shared structure between the representations, rather than arbitrary choices of units or target scaling.

### D.5 $I_\nu$ Mutual Information Plots

In addition to the probing results in Fig. 18, we investigate another probing approach with different mutual information variant. We reproduce the experiment in App. D via $\mathcal{I}_\nu\left(H_{1:T}^{1:L-1}; h_{T+\tau}^L\right)$ definition from Xu et al.. As it is seen from the results, Fig. 19, the $\mathcal{I}_\nu\left(H_{1:T}^{1:L-1}; h_{T+\tau}^L\right)$ definition is not helpful either. Also, we test validity of $\mathcal{I}_\nu$ with our easiest validation test from A.4.1. The results are seen in Fig. 20. As it is seen $\mathcal{I}_\nu$ fails on our validation test as well.

### E  Finite Sampling Error

We quantify the finite-sample error when estimating the mutual information between two discrete random variables $X$ and $Y$ over two different MI estimation approaches.

Let $X$ and $Y$ take values in $\{1, \ldots, K_X\}$ and $\{1, \ldots, K_Y\}$, with joint pmf $p_{ij} := \Pr(X = i, Y = j)$, marginals $p_i := \sum_j p_{ij}$ and $p_j := \sum_i p_{ij}$, and true mutual information

$$I(X; Y) = \sum_{i=1}^{K_X}\sum_{j=1}^{K_Y} p_{ij} \log \frac{p_{ij}}{p_i p_j}.$$

---

[4]Equivalently, $I_\nu(X \to Y)$ is the total variance of $Y$ times the fraction of variance explained by the best linear predictor.

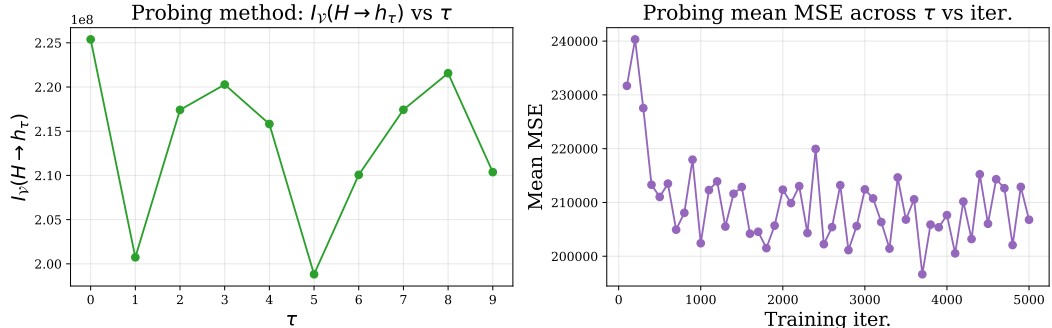

Figure 19: Reproduction of experiment in App. D via $\mathcal{I}_\nu$.

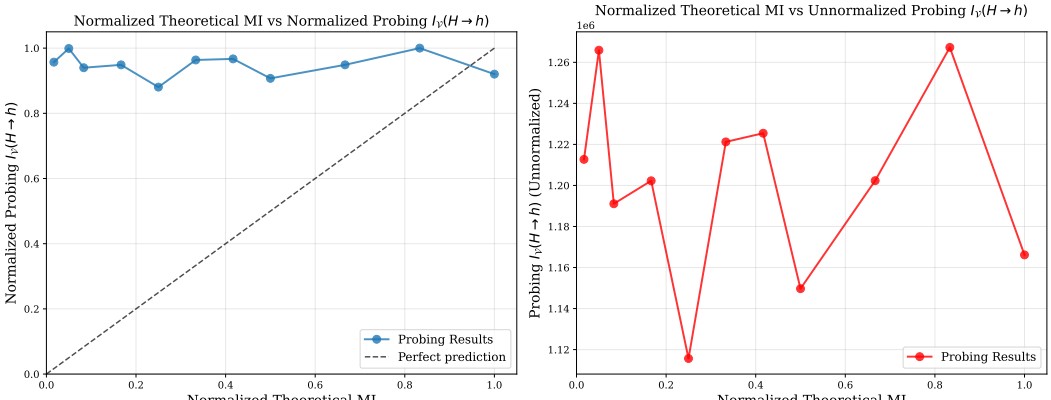

Figure 20: Plots to check validity of $\mathcal{I}_\nu$ on the easiest validation experiment in App. A.4.1

From $N$ i.i.d. samples $(X_t, Y_t)_{t=1}^N$ drawn from $(X, Y)$, define the cell counts

$$N_{ij} := \sum_{t=1}^N \mathbf{1}\{X_t = i, Y_t = j\}, \qquad N_i := \sum_{j=1}^{K_Y} N_{ij}, \quad N_j := \sum_{i=1}^{K_X} N_{ij},$$

and the empirical probabilities $\hat{p}_{ij} := N_{ij}/N$, $\hat{p}_i := N_i/N$, $\hat{p}_j := N_j/N$. The standard plug-in (maximum-likelihood) mutual information estimator is

$$\hat{I}(X; Y) := \sum_{i=1}^{K_X} \sum_{j=1}^{K_Y} \hat{p}_{ij} \log \frac{\hat{p}_{ij}}{\hat{p}_i \hat{p}_j}. \tag{6}$$

**Asymptotic bias and variance of the plug-in MI.** Under standard regularity assumptions (in particular, $p_{ij} > 0$ for all $i, j$, which are satisfied in our experiments), an exact local expansion plus a delta-method argument gives the following large-$N$ behavior (Panzeri & Treves, 1996; Treves & Panzeri, 1995; Paninski, 2003):

$$\mathrm{bias}\big(\hat{I}(X; Y)\big) := \mathbb{E}[\hat{I}(X; Y)] - I(X; Y)$$
$$= -\frac{(K_X - 1)(K_Y - 1)}{2N \ln 2} + O\left(\frac{1}{N^2}\right), \tag{7}$$

$$\mathrm{Var}\big(\hat{I}(X; Y)\big) = \frac{C_{\mathrm{var}}(X, Y)}{N} + O\left(\frac{1}{N^2}\right), \tag{8}$$

where $C_{\mathrm{var}}(X, Y) > 0$ is a constant that depends only on the true joint distribution $p_{ij}$. Consequently, the mean-squared error (MSE) satisfies

$$\mathrm{MSE}\big(\hat{I}(X; Y)\big) := \mathbb{E}\Big[(\hat{I}(X; Y) - I(X; Y))^2\Big] = \frac{C_{\mathrm{MSE}}(X, Y)}{N} + O\left(\frac{1}{N^2}\right), \tag{9}$$

for another constant $C_{\mathrm{MSE}}(X,Y) > 0$. In other words, the plug-in MI estimator has

$$\text{bias} = O\left(\frac{1}{N}\right), \qquad \text{Var} = O\left(\frac{1}{N}\right), \qquad \text{MSE} = O\left(\frac{1}{N}\right).$$

A simple first-order bias correction that cancels the leading $O(1/N)$ term in (7) is

$$\hat{I}_{\mathrm{PT}}(X;Y) := \hat{I}(X;Y) + \frac{(K_X - 1)(K_Y - 1)}{2N \ln 2}. \tag{10}$$

Then

$$\text{bias}\big(\hat{I}_{\mathrm{PT}}(X;Y)\big) = O\left(\frac{1}{N^2}\right), \tag{11}$$

$$\text{Var}\big(\hat{I}_{\mathrm{PT}}(X;Y)\big) = \frac{C_{\mathrm{var}}(X,Y)}{N} + O\left(\frac{1}{N^2}\right), \tag{12}$$

$$\text{MSE}\big(\hat{I}_{\mathrm{PT}}(X;Y)\big) = \frac{\tilde{C}_{\mathrm{MSE}}(X,Y)}{N} + O\left(\frac{1}{N^2}\right), \tag{13}$$

i.e., the MSE still scales as $O(1/N)$ but with a smaller constant prefactor (Treves & Panzeri, 1995).

**Paninski-type estimators and worst-case MSE upper bounds.** Paninski (2003) constructed a family of linear estimators for discrete entropy, and hence for mutual information via $I(X;Y) = H(X) + H(Y) - H(X,Y)$, by explicitly optimizing a rigorous upper bound on the worst-case MSE over all discrete distributions with a given alphabet size. The resulting "best universal bound" (BUB) estimators $\hat{I}_{\mathrm{BUB}}(X;Y)$ satisfy bounds of the form

$$\sup_p \mathbb{E}\Big[(\hat{I}_{\mathrm{BUB}}(X;Y) - I(X;Y))^2\Big] \le \frac{C_{\mathrm{BUB}}(K_X, K_Y, N)}{N}, \tag{14}$$

for an explicit constant $C_{\mathrm{BUB}}(K_X, K_Y, N)$ that can be computed from $(K_X, K_Y, N)$; in particular, the worst-case MSE is also upper bounded by a constant times $1/N$.

**What we use in this work.** In our experiments, we computed mutual information using both (i) the plug-in estimator (6), typically with the Panzeri–Treves bias correction (10), and (ii) Paninski's BUB-type estimators assembled from the corresponding entropy estimators (Paninski, 2003). In all cases we tested, both approaches produced essentially identical values of $I(X;Y)$. Since our available sample sizes $N$ are very large compared to the effective domain sizes $K_X$ and $K_Y$ (we have easy access to CFG, Path Finding, and NLP samples), the $O(1/N)$ finite-sample errors of both methods are very small. In most of our experiments we therefore report the mutual information estimated using the plug-in estimator.

## F  THE USE OF LARGE LANGUAGE MODELS (LLMs)

We used LLMs in the following ways.

- We polished sentences and corrected the grammar.
- For visualization (figures, plots) of the results, we made LLM to write the visualization code.
- For the experiment codes we got LLM help, e.g., tab completions, function generation given the prompt, and we checked all LLM generated code line by line to ensure it works as intended.
- At the beginning of the project, we used an LLM to help identify relevant literature and read the suggested papers most pertinent to our work.

