# OpenReview forum: "Internal Planning in Language Models: Characterizing Horizon and Branch Awareness"
_ICLR.cc/2026/Conference — ICLR 2026 Poster_

### Official Review · Reviewer_rbc9 · 2025-10-26

**Soundness:** 3
**Presentation:** 2
**Contribution:** 2
**Rating:** 6
**Confidence:** 3

**Summary:**

This paper investigates whether decoder-only language models (LMs) engage in internal planning, defined along three axes: forward-looking (planning beyond the next token), branch-aware (considering multiple valid continuations), and stateful (reusing earlier computations). The authors propose an information-theoretic framework that compresses high-dimensional hidden states into discrete codes using a modified Vector-Quantized Variational Autoencoder (VQ-VAE), enabling mutual information (MI) estimation between different parts of the model’s internal computations. They apply this pipeline across synthetic (CFG, path-finding) and natural language (OpenWebText) tasks, finding that planning behavior is task-dependent. Stronger evidence of non-myopic computation is observed in structured reasoning tasks than in syntactic or natural language settings.

**Strengths:**

- The paper is well-motivated. Previous studies on the internal mechanisms of LMs often rely on linear probing, which is susceptible to confounders and may lead to unreliable conclusions. The proposed method avoids these confounders, making the analysis of internal planning in LMs more reliable.
- The use of mutual information over compressed discrete codes provides a principled and scalable approach to analyzing high-dimensional activations.
- The experimental design covers a thoughtful range of tasks—from highly structured symbolic problems to natural language—allowing for nuanced insights into when and how planning emerges.
- The validation experiment in Appendix A.3 reasonably demonstrates that the VQ-VAE compression preserves meaningful statistical dependencies.

**Weaknesses:**

- Overall, the empirical findings are somewhat incremental. For instance, the observation that LMs plan more in path-finding tasks and exhibit myopic behavior in CFG tasks is somewhat expected. Additionally, the proposed framework does not offer a mechanistic explanation of how such planning is implemented, which limits the significance of the findings.
- In my view, using VAE-based methods also has drawbacks compared to linear probing:
  - Interpretability is relatively poor. Since the latent space of VAEs is inherently difficult to interpret, even if changes in MI are detected, it remains challenging to explain the underlying mechanisms behind these changes.
  - The VQ-VAE introduces its own set of hyperparameters and design choices (e.g., codebook size, cosine penalty), and the sensitivity of the results to these choices is not thoroughly explored.
- The reliance on normalized MI makes it difficult to assess the absolute strength of planning signals. Moreover, the paper does not convincingly demonstrate that the observed MI reflects causal planning rather than passive statistical correlation.

**Questions:**

- I am somewhat confused about the authors’ definitions of planning properties: "Stateful" involves reusing earlier computations, but isn’t this also a manifestation of being "forward-looking"? Could the authors provide more intuitive examples to clarify the distinction between these two concepts?
- Regarding the use of VAE for compressing information to obtain distributions:
  - Could the authors briefly explain why this method is unlikely to introduce external information? Is it because the two positions being compared are compressed by two independent VAEs? Is it possible that the learned encoder and codebook inadvertently encode external information, leading to misleading conclusions?
  - Could the same MI patterns arise in a model that does not "plan" in any meaningful sense?
- Based on the observed patterns of LM planning across different tasks, could the authors propose any improvements to the LM architecture or algorithms, such as a blueprint for enhancement?
- If earlier blocks retain "nontrivial information," as claimed, why doesn’t ablating or perturbing them significantly degrade performance? Such a test could potentially strengthen the argument for statefulness.

---

> ### Author Response · Authors · 2025-11-20
>
> We thank the reviewer for highlighting the strengths of our motivation, methodology, and experimental design. Below, **we respond to all of the reviewer’s concerns and questions in detail**:
>
> **[Whether the results are surprising and mechanistic explanation]**: We would like to clarify that our goal in the paper is not to claim surprising phenomena. We introduce an information-theoretic framework to study planning in LMs and use it to answer three important questions: horizon, branching, and history of the plan. Considering this, the fact that our findings is consistent with prior intuitions serves as a further validation of our method. Furthermore, for instance, our results show that using MTP over NTP does not significaly help to increase in "planning horizon" but underlying task(dataset) on which the model is trained is very important. Clearly, this finding both puts claims against MTPs' abilities and partly explains why we do not see significant multi token prediction on the modern very large language models.
>
> Regarding mechanistic explanation, we agree that our approach does not give instance-wise circuit explanations. Compared to labour-intensive mechanistic interpretability works [1, 2, 3], our approach offers a scalable, probe-free perspective on information flow without manual circuit tracing needed. Furthermore, we would like to note that mechanistic explanation-based studies can benefit our method to obtain high-level code-based summaries of internal computations, which can be an interesting future direction but out of scope of our current paper.
>
> [1] Elhage et. al, A mathematical framework for transformer circuits, 2021 \
> [2] Wang et. al, Interpretability in the wild: A circuit for indirect object identification in GPT-2 small, 2021 \
> [3] Bricken et al., Towards Monosemanticity: Decomposing language models with dictionary learning, 2023
>
>
> **[Revievers' concern on our method vs. linear probing]**:
> We fully agree that our pipeline can have drawbacks compared to linear probing, and for this reason we do not claim in the paper that it should replace linear probing across all interpretability research. However, we respectfully disagree with the reviewer regarding the suggested “interpretability drawback.” We would like to emphasize that linear probing is itself only a tool; it does not inherently provide interpretability. Interpretability arises when one carefully chooses what to probe and frames meaningful questions about the underlying model.
>
> We believe that nearly all meaningful interpretations obtainable through linear probes could also be derived through carefully designed experiments using our pipeline (while reiterating that we are not claiming our approach should entirely replace linear probing). The fundamental reason for our belief is simple: linear probing amounts to training a small model between thoughtfully chosen variables to test whether one variable contains enough information to predict another. Interpretability stems from these intelligent choices. Our approach can likewise assess whether one variable contains information about another, and in many cases may be even more suitable for interpretability-focused questions.
>
> Finally, we would like to highlight that interpreting the discrete latent space produced by our method presents a promising direction for future work.
>
> As for, the VQ-VAE's new set of hyperparameters, we are thankful to the reviewer for mentioning on them. Although we shared the hyperparameters in the provided code, we agree that it would be better to explain general intution about them in the paper. We therefore added a new section to the paper (see Appendix A.3) where we explain the sensitivity of the results to the coice of the hyperparameters.
>
> **[Can MI measure planning?]** Thank you for the thoughtful concern. We agree that MI gives us statistical relations, not a measure of planning; but this is not what we do. We do not treat MI as a direct measure of planning or causality.  Our use of MI is strictly comparative: we compute MI between high-level codes and interpret patterns across time indices, layers, and conditions to assess where and when information about future tokens appears, not to assign an absolute “planning score.” Normalization of MI supports this comparative use because each analysis is symmetric: the same model, encoder, codebook, and estimator are fixed within a setting; and we normalize by a common $I_{\text{max}}$ so that relative differences reflect the difference of how much computation a language model spares for future tokens comparatively. In the horizon study, for example, a sharp decay across $\tau$ indicates myopic computation, whereas sustained MI indicates forward-looking structure; the claim is about the trend, not its absolute magnitude. This also exemplifies how we do not claim a relation between MI strength and planning, but comperative interpretation. We clarify this in the updated manuscript as well.

---

> ### Author Response · Authors · 2025-11-20
>
> _**Replies to the Questions:**_
>
> **[Distinction between forward-looking and statefulness (history) experiments and improvement suggestions on existing LM methods based on our paper]**
>
> In short, 'forward-looking' experiment measures whether the current prefix computation already contains an implicit outline of future decisions, while 'history' (stateful) experiment measures how much the current decision depends on earlier computations in the context. We agree that, future-looking and history-looking are conceptually similar ideas. However, we want to underline/remind that our "planning horizon" and "history of the plan" experiments are distinct. The planning horizon experiment looks at how much preoutput computation (so all hidden states for the preoutput computation) contains information about future decisions. However, the "history of the planning" experiment mostly looks at which old calculations (which layer and time blocks) are more crucial for the current decision.
>
> That brings us to the question whether "the authors could propose any improvements to the LM". Using the results of the 3rd experiment we can provide a guide for how one should vary the attention window size over different layers of the language model. Although there are some rule of thumb preferences in the literature (as prefer longer attention windows for last layers and smaller windows for earlier layers), deciding the exact window size for each layer still depends on trial and error. Unfortunately, the combinations of attention window settings across layers grows polynomially with the window size, so the combinations to try in a loose sense would grow $O({T}^{\text{c} \times L})$, where $c$ is just a constant. One can also decide on the sizes according to the actual utilizations of windows, shortly, spare longer attention window for the layers in which you see history dependence persists for older blocks and vice versa. Luckily, the computational complexity of the **all** history experiment grow linearly with the attention window.
>
> However we want to note that the methodotical improvement blueprints and verifications are beyond the scope of this work. We believe that in the future following research could be to come up blueprint using our approach and verify the results on practical language models by comparing them on benchmarks.
>
>
> **[Regarding the use of VAE for compressing information to obtain distributions]**
> First, we explain why this approach cannot introduce extra mutual information, using the “planning horizon” experiment as an example (though the argument easily generalizes to other settings). Assume we separate the pre-output computations into two random variables, $H = (h, h’)$, where $h$ contains the parts of the computation that carry information about the generated token, and $h’$ contains the parts that do not. We then quantize $H$ using a VQ-VAE and obtain a code $Z$. This code can similarly be written as $Z = (z, z’)$, where $z$ is the component informative about future tokens, and $z’$ is the component that is not informative about future tokens. Although it may appear that the VQ-VAE introduces extra information via $z’$ (i.e., information irrelevant to the future tokens), $z’$ does not affect the mutual information between $Z$ and the future tokens. Therefore, the VQ-VAE cannot introduce additional mutual information.
>
> As for whether similar MI patterns could appear in a model without meaningful planning, our results suggest that the patterns we measure are closely tied to properties of the specific model under study rather than being generic artifacts. In the planning-horizon experiment, for example, the MI profiles change systematically across models with different demonstrated planning abilities. This variation indicates that the patterns are not some vague byproduct of the analysis pipeline, but instead track differences in how the models actually process and retain information.
>
>
> **[About ablating and perturbing analysis]** Thank you for pointing out perturbation analysis. Just to make clear, we do not claim that perturbing earlier blocks would leave performance unaffected. Since our results indicate that these blocks retain nontrivial information, perturbing them would likely cause a meaningful drop in performance. Thus, our results already confirm the possible alternative investigation you suggest. Also, we think a careful perturbation analysis would be a cool follow up research direction.
>
> We hope our responses resolve the concerns and address the questions. We would be **happy to clarify anything further**, and with this additional context and clarity, we **appreciate your consideration in raising the score**.

---

### Official Review · Reviewer_cc54 · 2025-10-26

**Soundness:** 2
**Presentation:** 2
**Contribution:** 2
**Rating:** 4
**Confidence:** 3

**Summary:**

For interpretability purposes, it is useful to be able to estimate the mutual information (MI) between two distinct blocks of activations in a transformer model. The authors propose to do this by quantizing these activations using a specially trained Vector-Quantized Variational Autoencoder (VQ-VAE), then measuring the MI between the resulting discrete codes. This novel interpretability technique is then used to measure the presence of planning in transformer models in both toy settings and natural language.

**Strengths:**

- As far as I'm aware, the use of VQ-VAE to estimate the mutual information between high-dimensional continuous activations is novel.
- Several distinct aspects of planning are investigated in both toy settings and natural language.
- The authors acknowledge that their absolute MI estimates may be misleading, and focus on relative comparisons.
- The authors provide a basic sanity check to validate their MI via VQ-VAE technique (A.3).

**Weaknesses:**

1. I am not fully convinced by the authors' philosophical argument that MI via VQ-VAE is more appropriate than existing linear probe techniques.
	- To fix notation, say that we care about two model activations $h_1,h_2\in\mathbb{R}^d$. We can either train two VQ-VAE encoders $E_1,E_2$ and estimate the mutual information $I(E_1(h_1);E_2(h_2))$, or we can train a probe $\phi\colon\mathbb{R}^d\to\mathbb{R}^d$ and measure the $\ell_2$ loss $||\phi(h_1)-h_2||_2^2$.
	- My understanding is that *both* of these, up to some constant factor, are valid lower bounds for the quantity of interest $I(h_1; h_2)$. The former is a lower bound by the data processing inequality, while the latter is by [1, Prop 1.5].
	- The extent to which these bounds are loose is due in the former case to the information discarded by $E_1,E_2$, and in the latter case to the inability of the probe architecture to represent the true conditional probability $P(h_2\mid h_1)$. It's not clear to me why the former approach is better or more principled than the latter.
	- I'm not sure I understand what is meant by the "confounding effect" of learned probes mentioned in the introduction. Both the VQ-VAE approach and the probing approach require training separate auxiliary models, and the quality of the lower bound depends on the representational capacity of the auxiliary model in either case.
	- I would argue that in the neural network setting, *both* approaches are somewhat philosophically dubious. It is often the case that $h_2$ is a deterministic function of $h_1$ (for example, in the setting of Section 3.1, where $h_1=h_{1:T}^{1:L-1},h_2=h_{T+\tau}^L$, this is true if sampling temperature is zero). In this case $I(h_1;h_2)=H(h_2)$, telling you nothing about the relationship between $h_1$ and $h_2$.
2. Experimental results have no comparisons with baselines such as linear probes.
	- Since the theoretical/philosophical discussion is somewhat unclear, an empirical comparison would give much stronger evidence for the suitability of MI via VQ-VAE vs probes for interpretability.
	- As it stands, the paper applies an unproven method to novel toy settings without established baselines. Without proper baselines, it is hard to be confident that these results are correct or meaningful.
3. MI between past and future model activations is a fairly weak information-theoretic condition, and is e.g. always present in the zero-temperature autoregressive setting. In my opinion, "planning" is a somewhat misleading term for this, as it implies intentionality in the model which may not really be present.

 [1] Xu et al. "A theory of usable information under computational constraints". ICLR 2020.

**Questions:**

- How canonical are the three aspects of planning you study here (forward-looking, branch-aware, stateful)? Are you claiming that possessing all three properties is either necessary or sufficient for being a good planner?
- One advantage of linear probes is that they are cheap and simple to implement. How expensive is the VQ-VAE to train? Does it scale to the kinds of LLMs that may be used in practice (billions of parameters)?
- What temperature was used for sampling in the experiments?
- I strongly suggest adding a comparison to linear probe baselines. I.e., wherever $I(E_1(h_1), E_2(h_2))$ is measured, instead fit a linear probe from $h_1$ to $h_2$ and report $R^2$.
- In the introduction, I recommend fleshing out the discussion of MI vs probes and citing [1].
- In "training objectives: next-token vs multi-token", I recommend citing [2], which shows that cross-token gradients induce forward planning even with a next-token objective.
- I like the sanity check performed in A.3, but having a discrete domain with support only 0-1 OOMs larger than codebook size is maybe too artificial. Are there continuous distributions for which the theoretical MI can be computed and compared with the estimated MI via VQ-VAE? Alternatively, what if the support is exponentially larger than codebook size (which is what happens in practice, because there are exponentially many possible input sequences)?

[1] Xu et al. "A theory of usable information under computational constraints". ICLR 2020.

[2] Wu et al. "Do language models plan ahead for future tokens?" COLM 2024.

---

> ### Author Response · Authors · 2025-11-20
>
> We thank the reviewer for their constructive feedback and for recognizing the novelty of our method. Below, **we respond to all of the reviewer’s concerns and address each question** in detail.
>
> **[Clarification regarding our technique and comparison with probing]**: We thank the reviewer for their questions and paper suggestion from the literature. We would like to answer this question in parts below:
>
> **1a. [Clarifying the suggested paper's results (Xu et al., 2020)]**
> We appreciate the reviewer’s thoughtful interpretation, but we respectfully disagree with the reviewer on the claim that Prop 1.5 in Xu et al., 2020 says with upper bounding $\| \phi(h_1) - h_2 \|^2$, we have a lower bound for "the quantity of interest $I(h_1; h_2)$". The **squared error of a linear probe $\phi$ is not, in general, a lower bound on Shannon mutual information. The result cited by the reviewer (Proposition 1.5 in Xu et al., 2020) does not identify the probe loss with $I(h_1;h_2)$.** Instead, for a restricted Gaussian linear family $\mathcal{V}$, it shows that $I_{\mathcal{V}}(X\to Y) = \text{tr}(\mathrm{Cov}(Y)) \, R^2(X\to Y)$, where $I_{\mathcal{V}}$ is the predictive V-information and $R^2$ is the coefficient of determination of the best linear predictor of Y from X. Thus the probe’s optimal mean–squared error is an affine transform of $I_{\mathcal{V}}(X\to Y)$, not of the Shannon mutual information I(X;Y). **Indeed, a key point of Xu et al. is that $I_{\mathcal{V}}$ can violate data processing and can increase under preprocessing, whereas I(X;Y) cannot**.
>
> **1b. [On the broader issue of linear probing and why it is not an alternative to our method]** Our paper does not aim to criticize probing-based analyses and indeed, there can be settings where probing may be preferable. However, probing is not the natural tool for the phenomenon that we study in our experiments. This insight aligns with a substantial body of prior work (some of which we cite in our literature review) that highlights conceptual limitations of linear probing as a measure of information content. In fact, these limitations mirror the behavior revealed by Xu et al. (2020) (the suggested paper). Even the $\mathcal{V}$-information $I_{\mathcal{V}}(X\to Y)$ is driven largely by the marginal complexity of the target $Y$: its scale is explicitly proportional to $\mathrm{tr}(\mathrm{Cov}(Y))$.
>
> This creates well-known confounding effects. For example, if $Y$ is (nearly) constant, a linear probe can achieve perfect prediction accuracy, yet both the true mutual information $I(X;Y)$ and the $\mathcal{V}$-information $I_{\mathcal{V}}(X\to Y)$ collapse to zero because $\mathrm{tr}(\mathrm{Cov}(Y))\approx 0$. Conversely, simply rescaling $Y$ via $Y \mapsto \alpha Y$ multiplies $I_{\mathcal{V}}$ by $\alpha^{2}$ without altering the underlying Shannon mutual information.
>
> Thus, there is no universal “constant factor’’ relating probe loss or $R^{2}$ to $I(h_{1};h_{2})$: the relationship is inherently data-dependent and can vary arbitrarily across tasks, distributions, and representation scales. This sensitivity is exactly the type of confounding issue that earlier probing-critique papers emphasize.
>
> Nevertheless, we fully agree that our first quantity, the mutual information between compressed codes $E_1(h_1)$ and $E_2(h_2)$ is a genuine lower bound on the Shannon mutual information $I(h_1;h_2)$ by data processing inequality. While it is also true that one can, in principle, obtain loose lower bounds on $I(h_1;h_2)$ (or in general $I(X, Y)$ from prediction error (e.g., via Fano-type inequalities applied to an appropriate discretization of $Y$), such bounds depend sensitively on how $Y$ is discretized and scaled. As a result, they do not yield a stable or task-independent notion of information suitable for representation-level analysis. Pursuing this direction would constitute a substantially different line of investigation, and falls outside the scope of the present work.

---

> ### Author Response · Authors · 2025-11-20
>
> **1c. [Simple setting on the confounding effects of learned probes]**
>
> We thank for the reviewer for pointing out the possible confusion in this issue. Here we explain why we respectfully disagree with the reviewer on "both VQVAE approach and probing approach's quality depends on the representational capacity of the auxiliary model". Because this point is important, we have added a dedicated clarification in the revised manuscript (Appendix D.2 and D.3), where we explain the issue in greater detail.
>
> To illustrate the “confounding effect’’ we refer to, we provide a simple example in which probing-based methods fail to reflect the true information-theoretic relationship between variables. Consider the following random variables. Let $X \sim \mathrm{Unif}[0,10]$ so that $H(X) = 3.32$ bits. Define two targets:
> $$
> Y_1 = 0 \quad\text{(a constant)}, \qquad
> Y_2 =
> \begin{cases}
> X, & \text{with probability } 0.5;
> 0, & \text{with probability } 0.5.
> \end{cases}
> $$
> It is clear that $X$ contains **strictly more information** about $Y_2$ than about $Y_1$. Indeed:
> - $I(X;Y_1)=0$ since $Y_1$ is constant.
> - $I(X;Y_2)=H(X) - H(X \mid Y_2)=1.66$ bits.
>
> This is summarized below:
>
> | Measure | $Y_1$ | $Y_2$ |
> |--------|-------|--------|
> | **$I(X;Y_k) = H(X)-H(X\mid Y_k)$** | $0$ | $1.66$ |
> | **Probe loss $\lVert \phi(X) - Y_k\rVert_2^2$** | $0$ | greater than zero |
>
> However, a probing model would achieve **zero error** on $Y_1$, because $Y_1$ is trivial to predict, while it would have much higher error on $Y_2$. A probe-based interpretation would therefore conclude incorrectly that *$X$ contains more information about $Y_1$ than about $Y_2$*. This directly contradicts the true mutual information values. Probe performance is strongly shaped by the **marginal complexity of the target**, not by the actual information flow from $X$ to $Y$. When the target is simple, low-variance, or nearly constant, probes can give misleadingly optimistic results; when the target is complex or multimodal, probes can dramatically underestimate the information present.
>
> For completeness, we note that our pipeline does not exhibit this pathology. Because we measure mutual information between *compressed representations* $E_1(h_1)$ and $E_2(h_2)$, rather than prediction accuracy, our estimates depend only on the shared structure between the variables and are invariant to trivial or low-variance targets. This ensures that, unlike probing, our method correctly reflects the underlying information-theoretic relationships in scenarios like the example above.

---

> ### Author Response · Authors · 2025-11-20
>
> **[Clarifying the use of “planning”]**: Thanks for raising this. We agree that MI captures statistical dependence. Our claim is not that MI measures planning itself. We comparatively use MI as a tool to understand where and when information about future tokens appears in the computation. Concretely, our analyses interpret planning comparatively. We learn discrete summaries of hidden states, compute MI between summaries tied to past and future variables, and then compare MI patterns across offsets, layers, or conditions. In the horizon study, for example, we examine how MI varies with the future offset $\tau$. The interpretation is about the pattern of change across $\tau$, not the absolute MI value. We clarify this in the updated manuscript as well.
>
>
> **[What happens if $h_2$ is a deterministic function of $h_1$]**: Thank you for pointing out a possible confusion. However we respectfully disagree with the reviewers' claim that "when the functional relation between $h_1$ and $h_2$ is deterministic, it tells nothing about the relationship between $h_1$ and $h_2$". Assuming $h_2$ is deterministic function of $h_1$, in our setting even if we had zero sampling temperature we would have $I(h1, h2) = H(h_2) < H(h1)$ in general. Even in this case, mutual information gives a useful interpretation about the planning. For instance in the "planning horizon" experiment on CFG dataset, we see a decrease in mutual information we look at further away generated decisions. That is, letting $h_3$ be distant last layer hidden state representation, we see $I(h1, h3) < I(h1, h2)$. That phenomenon is mainly based on the degree of injectivity of those functions, i.e., $h_3$ could be thought of as a coarser function of $h_1$ compared to $h_2$ is a function of $h_1$. To clarify, in our experiments, the sampling temperature is not zero, so the stated functional relations are not deterministic.
>
> _**Replies to the Questions:**_
>
> **[New probing experiments]**: We thank the reviewer for the suggestion. Following this, we added probing results in the updated Appendix D.2 and D.3. We evaluated both token-prediction and hidden-state regression probes using the same setting as our larger-model experiment. Token probes were only informative at immediate next token, while regression probes produced nearly flat normalized MSE across future offsets. Even with nonlinear two-layer MLP probes, the results did not reveal meaningful horizon structure or patterns relevant to planning. In contrast, our MI analysis extracts clear comparative trends across
> $\tau$. We emphasize that probing and our method answer different questions: probes test supervised recoverability of specific targets, whereas our framework measures unsupervised statistical dependence between compressed representations and we interpret the planning behaviour of LM by comparing these statistical measures across variables. For more detail, please refer to Appendix D.2 and D.3 in the updated manuscript.

---

> ### Author Response · Authors · 2025-11-20
>
> **[Validation Experiment (Sanity Check)]**: Thanks for the thoughtful question. We chose a discrete data-generating process because our primary use case is transformer LMs with discrete tokens as in the practice whose computations are then mapped into continuous hidden states. Appendix A.3 builds a controlled setting where the ground-truth dependence is known, letting us check whether VQ-VAE codes recover the intended information relations. Our proposed method in the paper targets this transformers/LM regime rather than arbitrary continuous distributions, which is why the original sanity check used discrete support. We **add two new validation experiments based on the reviewer’s suggestions**:
>
> - **Validation experiment, harder setting with multiple similar prefixes for each label (updated App. A.3.2)**: We create a more challenging variant of our existing validation experiment in A.3.1. For each $a \in A$ we form a set of 16 surrogate hidden-state tensors in $\mathbb{R}^{10 \times 11 \times 768}$: start from a random length-10 prefix, then make 15 additional prefixes by changing one token at a random position, and run the trained LM to collect the hidden state blocks. For each $b$ we similarly produce 16 final-layer states in $\mathbb{R}^{768}$ via one-token perturbations. The experiment therefore has $2^{14}$ possible $G_A$ blocks and $2^{14}$ possible $G_B$ targets **where our codebook sizes are much smaller**. This mimics text inputs that differ at one token yet keep nearly the same semantics, which our coarse codes should compress together. **To further push toward a continuous-like mapped distribution as suggested by the reviewer**, we add Gaussian noise during VQ-VAE training on hidden states. We compute normalized MI between learned coarse variables, treating the 16 surrogates for a fixed label as equiprobable. Result: across all codebook sizes $K$, the codes preserve the ground-truth ordering across $p_{\text{valid}}$, and $\text{nMI}(Z_A; Z_B)$ reflects the reference mutual information closer as $K$ increases.
>
> - **Validation experiment, hardest setting with multiple unrelated prefixes for each label (updated App. A.3.3)**: We now remove within-class similarity entirely. For each $a \in A$ we sample 16 independent length-10 token sequences uniformly at random and collect their $\mathbb{R}^{10 \times 11 \times 768}$ hidden-state tensors; for each $b$ we do the same for the final-layer $\mathbb{R}^{768}$ states. The 16 surrogates that share a label need not be close in vector space, emulating many-to-one mappings from highly diverse inputs to the same high-level code, similar to natural language where dissimilar texts can share meaning. Again there are $2^{14}$ possible $G_A$ and $2^{14}$ possible $G_B$. We repeat the pipeline and probability treatment from A.3.2 and estimate normalized MI with equiprobable surrogates. Result: even with 16 equally likely outcomes per input and no local similarity to exploit, our method preserves the correct ordering across $p_{\text{valid}}$, with improved agreement at larger $K$. We would like to note that **this regime is particularly difficult for linear probes, which would struggle under many-to-one, equally-probable unrelated targets**.
>
> These additional test settings **have much larger effective support and with continuous-like mapped distributions as asked by the reviewer**. In both, the VQ-VAE codes maintain the intended ordering of dependencies, supporting our claim that normalized MI over coarse summaries give meaningful and interpretable results.

---

> ### Author Response · Authors · 2025-11-20
>
> **[How canonical are the three aspects of planning, are you claiming necessity or sufficiency?]** To clarify, we are not presenting forward-looking, branch-aware, and stateful behavior as a complete or canonical definition of planning, nor do we claim that having all three would be sufficient for good planning. We chose these aspects because they recur across much of the planning literature—for example, forward simulation and multi-branch reasoning in classical AI and model-based RL (e.g., Sutton & Barto; Russell & Norvig), and explicit branching in recent MCTS approaches (AlphaZero, MuZero) and LM-based approaches such as Tree-of-Thoughts, Graph-of-Thoughts, LATS. They are therefore well-motivated, but certainly not exhaustive.
>
> We also do not intend to suggest that all three properties are strictly necessary. In particular, “statefulness” is better viewed as something that tends to make planning more efficient and robust rather than an absolute requirement. Other elements, such as exploration strategies, credit assignment, and model accuracy, also play important roles in effective planning.
>
>
> **[Remaining questions]** We use the sampling temperature between 0.5-1 depending on the experimental setting.
> We have discussed and cited the mentioned papers. For the scalability of our method in larger language models, we have added a new experiment in Appendix D.1 in addition to our previous scaling experiment in Appendix B.6. For more details of these experiments and results, please refer to our "[Experiment with larger models and scale]" to Reviewer 2.
>
> We hope our responses resolve the concerns and address the questions. We would be **happy to clarify anything further**, and with this additional context and clarity, we **appreciate your consideration in raising the score**.

---

> > ### Comment · Reviewer_cc54 · 2025-11-24
> > **Response to rebuttal (1/2)**
> >
> > Thank you to the authors for the detailed response and additional experiments. I have a few follow-up remarks:
> > - My initial review was indeed too hastily written, and made a couple of theoretical mistakes. I thank the authors for pointing them out:
> > 	1.  With the Gaussian linear family $\mathcal{V}$ defined as in Prop 1.5 of Xu et al., the expression for $I_{\mathcal V}(X\to Y)$ is $\text{tr }\text{cov}(Y) R^2$. When comparing $I_{\mathcal V}$ across multiple $Y$'s with differing variances (as is the case in the authors' setting), the covariance trace is not just a constant factor times $R^2$, as I had erroneously claimed. On the other hand, this itself is not a major technical limitation of probes, since it is cheap and straightforward to estimate the empirical $\text{tr }\text{cov}(Y)$ and multiply it in. (I agree with the authors that there are other reasons why $I_\mathcal V(X\to Y)$ might be a poor estimate of $I(X;Y)$, which I discuss below.)
> > 	2. Although conditional $\mathcal{V}$-entropy is an upper-bound on standard conditional entropy (following directly from the infimum definition of $\mathcal{V}$-entropy), there is in general no inequality between standard and $\mathcal{V}$-mutual information; this is because the mutual information is the difference between two (conditional) entropies. However, **one can still informally understand $I_{\mathcal V}$ as an approximation of $I$ that becomes exact as $\mathcal{V}$ becomes more expressive** (using the notation in Sec 2 of Xu et al.) To be sure, this is not the same as saying $I_{\mathcal V}\to I$ as the *probe* becomes more expressive, because in the setting of Prop 1.5 the probe controls only the mean of an isotropic Gaussian whose covariance is held fixed.
> > 		- As an aside, it is also the case that $I_\mathcal{V}(X\to Y)$ lower bounds $I(X;Y)$ when the the law of $Y$ is contained in $\\{f(\emptyset)\mid f\in \mathcal{V}\\}$, which, in our setting, corresponds to $Y$ being an isotropic Gaussian with fixed variance and arbitrary mean. I agree that this is not a realistic assumption to make in the authors' setting.
> > - Following (1.) above, I believe the correct probing baseline to use in an activation-regression setting is $\text{tr }\text{cov}(Y) R^2$, which can be directly interpreted as an informal approximation of the mutual information. As the authors point out, this measure fails to satisfy the data-processing inequality in a somewhat egregious way (rescaling $Y$ also rescales the measure), so this baseline is likely significantly outperformed by the author's proposed VQ-VAE MI. However, I think it's worth making the comparison explicit. Also, in the authors' confounding example (D.3), the quantity $I_{\mathcal V}$ gets the desired behavior $\text{tr }\text{cov}(Y_1) R_1^2< \text{tr }\text{cov}(Y_2) R_2^2$

---

> > > ### Comment · Reviewer_cc54 · 2025-11-24
> > > **Response to rebuttal (2/2)**
> > >
> > > **Suggested new experiments**
> > > - Plot $I_{\mathcal V}(X\to Y)=\text{tr }\text{cov}(Y) R^2$ instead of normalized MSE in the new probing experiments (Fig 18).
> > > - Although Fig 18 is suggestive that probes fail to find meaningful information structure that VQ-VAE finds, it is hard to be certain, because there is not a known ground truth for this setting. On the other hand, the validation experiment (Sec A.4) do have a ground truth. Plot probe $\text{tr }\text{cov}(Y) R^2$ for the validation experiments and make a direct comparison between the accuracies of $I_{\mathcal V}$ and VQ-VAE MI as estimates of the theoretical true MI.
> > > - I'd be interested to see $H(X)$ and $H(Y)$ included for the $I(X;Y)$ plots (e.g. Figs 3 (a) and 17). (The same $\max I(X;Y)$ denominator for normalizing $I(X;Y)$ should be used for $H(-)$.) To what extent is the trend wrt $\tau$ driven by decrease in $H(Y)$ vs increase in $H(Y\mid X)$?
> > >
> > > **Suggested writing changes**
> > > - Although Sec. D.3 explains what the authors mean when they state that their method is, unlike probes, "free from confounding", the distinction may be somewhat difficult to grasp from the introduction alone. Another approach is to mention in the "Contribution 1" paragraph that $I_{\mathcal V}$ from probes can also be thought of as informal approximations of MI, but that the authors' proposed approach has 1) nicer theoretical properties and 2) outperforms $I_\mathcal V$ from probing in both the toy validation task and the CFG task. Thus, instead of a somewhat philosophical discussion, there is a crisp apples-to-apples comparison for which VQ-VAE MI is better.
> > > - The "What should these results mean? Are they baselines?" paragraph can be changed/removed once $I_{\mathcal V}(X\to Y)=\text{tr }\text{cov}(Y) R^2$ is plotted instead of normalized MSE.
> > > - Personally, I found that the more important contribution of the paper is the novel VQ-VAE MI measure. The planning tasks are interesting as well, but are only one application (and a validation of) this tool. It might be better to re-structure the introduction to mention the VQ-VAE MI contribution prior to the long discussion of planning. This is largely a matter of taste, though, and I'll leave it to the authors' judgement.

---

> > > > ### Author Response · Authors · 2025-11-26
> > > > **[Reply to part 2/2]**
> > > >
> > > > **About Suggested New Experiments:**
> > > >
> > > > **[Specific $I_\nu = tr cov(Y) R^2$ probing plot]**
> > > > We thank the reviewer for the suggestion. In addition to the probing results in Fig. 18, we investigate another probing approach. We reproduce the "Planning Horizon" experiment via $I_\nu$ definition you suggested. As it is seen from the results Fig.21, it does not have meaninful results.
> > > >
> > > > **[Probing experiments on the Sanity Check task]**
> > > > We appreciate the reviewer’s suggestion to apply the same probing baseline to the sanity-check experiment. We implemented the $I_\nu$ probing approach in the simplest version of this validation task; the results are shown in Fig. 22. In this controlled setting, $I_\nu$ does not recover the ground-truth ordering, whereas our VQ–VAE MI estimator does. This outcome highlights that, in these particular experimental scenarios, MI between learned discrete representations provides a clearer and more reliable signal than probe-based quantities.
> > > >
> > > >
> > > > **[On plotting $H(X)$ and $H(Y)$ versus $\tau$.]**
> > > > We agree that, in principle, it is helpful to separate changes in $I(X;Y)$ into changes in $H(Y)$ versus $H(Y\mid X)$. In our setting, however, $H(Y)$ is essentially constant as a function of $\tau$. Intuitively, $Y$ is the VQ–VAE code summarizing the hidden state at offset $\tau$ in the continuation. For both our CFG data and the natural-language data, the marginal distribution over tokens (and hence over their hidden states and VQ–VAE codes) is translation-invariant across positions: there is no systematic asymmetry between “near-future” and “far-future” tokens if one does not restrict to a very specific positional subset. Consequently, the marginal entropy $H(Y)$ is not expected to vary with $\tau$.
> > > >
> > > > To verify this empirically, we have added an appendix G figure that plots the entropies of both code distributions, $H(VQ(h_{\text{prefix}}))$ and $H(VQ(h_\tau))$, for both small and large models. For both models $H(VQ(h_{\tau}))$ is nearly constant as expected. This confirms that the decreasing trend in our normalized $I(X;Y)$ curves with increasing $\tau$ is driven almost entirely by an increase in the conditional entropy $H(Y\mid X)$.
> > > >
> > > >
> > > > **About the suggested writing changes:**
> > > >
> > > > **[Strengthened Version of Contribution 1]**
> > > > Thank you for the suggestion. We agree that clarifying why our method is “free from confounding,” especially relative to probe-based measures, strengthens the framing of Contribution 1. As you recommended, we revised the Contribution 1 paragraph to make this distinction more explicit. However, to keep the Introduction accessible and avoid introducing technical notation such as $I_\nu$ or probe-specific loss formulations at an early stage, we opted for a higher-level explanation that highlights the core conceptual differences without relying on formal definitions.
> > > >
> > > > Specifically, the updated text (see Appendix F for now, however we will update the main text along this line for the camera ready version) now emphasizes that probe-based approaches rely on an externally trained predictor, whose representational capacity and objective can introduce confounding influences that obscure the underlying LM computation. In contrast, our method computes mutual information (MI) directly between learned discrete representations summarizing internal LM states. This provides a confound-resistant way to measure shared information without injecting additional modeling assumptions or auxiliary predictor capacity.
> > > >
> > > > **[Other writing suggestions]**
> > > > We thank the reviewer for these helpful suggestions. Regarding the paragraph “What should these results mean? Are they baselines?”, we agree that change indeed strengthens the stance of our method relative to probing, and we consider how to incorporate this clarification along with the last suggestion (the weighting for paper's contributions) in the camera-ready version, where we will also revisit that paragraph to streamline the exposition.
> > > >
> > > > At the same time, we would like to emphasize that our paper is not positioned as a refutation of probing or as a general-purpose improvement over probing methods. Probing plays an important role in the interpretability literature, and we cite prior critiques of probes purely as part of the broader context. Our contribution is centered on a different aim: we introduce an MI-based framework, enabled by VQ–VAE compression, which is particularly suitable for the for our planning experiments.
> > > >
> > > > **We thank the reviewer again for their detailed follow-up and for the suggestions. We have addressed the points raised, added the requested $I_\nu$ probing experiments, extended the sanity-check validations, included the $H(Y)$ analysis, and clarified the framing in Contribution 1. We hope these updates resolve the remaining concerns, and we appreciate the reviewer’s consideration in updating their evaluation**.

---

> > > > > ### Comment · Reviewer_cc54 · 2025-11-26
> > > > > **Response to new changes**
> > > > >
> > > > > Thank you for the detailed reply; I greatly appreciate the incorporation of new experiments and writing changes.
> > > > > I now agree that the VQ-VAE MI is a novel interpretability tool that cannot be replaced with simple probing techniques. Accordingly, **I've updated my score to a 6.**

---

> > > ### Author Response · Authors · 2025-11-26
> > > **[Reply to part 1/2]**
> > >
> > > **[On predictive $I_\nu$ and our Appendix D.3 example.]**
> > > We thank the reviewer for carefully revisiting the theory of Xu et al. and for confirming that our objections in the rebuttal were correct.
> > >
> > > Our toy construction in the original Appendix D.3 was intentionally formulated to illustrate a limitation of standard linear probing losses, which are by far the most commonly used quantities in the probing literature, rather than to target the specific $I_\nu$ definition of Xu et al. We deliberately kept this example as simple as possible so that the intuition is accessible to a broad audience, many of whom may not be familiar with predictive $\nu$-information.
> > >
> > > To address the specific $I_\nu$ definition, we have added Appendix D.4. which has a new example showing a pathology that is specific to the $I_\nu$ definition. In this new construction, two targets $Y_1$ and $Y_2$ are related by an invertible linear rescaling, so that $I(X;Y_1) = I(X;Y_2)$ in the Shannon sense, but $I_\nu(X \to Y_2) = a^2 I_\nu(X \to Y_1)$ can be made arbitrarily larger or smaller by changing the scale factor $a$. This concretely illustrates the reviewer’s (and our) concern that $I_\nu$ violates data processing and is highly sensitive to the marginal scale of $Y$.

---

> ### Comment · Reviewer_cc54 · 2025-11-26
> **Additional comments**
>
> I thought I should briefly remark as to my remaining uncertainties about the paper (and why not an even higher score):
> - While I now agree that the VQ-VAE MI is a substantive and novel interpretability tool, it seems more coarse-grained than others, perhaps limiting its usefulness. Probes and circuit-tracing style tools (e.g. [1]) can answer questions like:
>   1. What *specific* examples in the training corpus does the model plan either more or less on?
>   2. What does the model "think" the upcoming token's value should be? How does it arrive at that answer?
>
>   As far as I am aware, MI, while an interesting measure, cannot directly answer these questions.
> - The fact that a novel approach is being used on novel settings still makes me somewhat uncertain. Is there previous literature that attempts to measure planning in either the CFG or path-finding settings, and can direct comparisons be made to those papers? Alternatively, how does VQ-VAE MI do in settings where past work has claimed to find planning, e.g. the rhyming poetry example [1, Sec 4]?
>
> I will also note that it is possible I am underrating the paper's Contribution 2 (understanding LM planning), since I am unfamiliar with the MPC literature that the authors cite.
>
> [1] "On the Biology of a Large Language Model." Anthropic.

---

> > ### Author Response · Authors · 2025-11-27
> >
> > Thank you for thoughtful reconsideration and raising the score. We are grateful for your careful reading and constructive feedback. We would like to answer your remaining questions.
> >
> > **[(1) “What specific examples in the training corpus does the model plan either more or less on?”]**
> >
> > Our method can help this by computing sample-wise mutual information, rather than aggregating MI over the entire dataset. Given two hidden-state sequences $h_1, h_2$ and their VQ-VAE encodings $E_1(h_1), E_2(h_2)$, the sample-wise MI for a single example $x$ is
> >
> > $$I_x(h_1; h_2) \;=\; \log \frac{p(E_1(h_1(x)),\, E_2(h_2(x)))}{p(E_1(h_1(x)))\,p(E_2(h_2(x)))}$$
> >
> > Using this pointwise quantity (or by choosing any favorite pointwise MI variant), we can  inspect which specific training samples exhibit stronger or weaker planning behavior. Since our method already produces discrete codes for each sample, we can compute $I_x(h_1; h_2)$, and we believe this approach can be extended and utilized in works exploring sample specific relations similar to [1].
> >
> >
> > **[(2) “What does the model ‘think’ the upcoming token’s value should be? How does it arrive at that answer?”]**
> >
> > Answering the explicit content of the model’s token-level prediction requires tools beyond pure MI, as MI quantifies shared information rather than the form of the prediction itself. However, our approach can be combined with fine-grained interpretability methods to obtain a richer picture. For instance, one could extend our method with multi-label VQ-VAE. Applying MI to these structured latent variables would reveal which components of the representation contribute to planning and how these components support the model’s prediction mechanism. Hybridizing our approach with circuit-tracing techniques provides a promising path toward explanations of why the model favors a particular next token.
> >
> >
> > **[“Is there previous literature that attempts to measure planning in the CFG or path-finding settings, and can direct comparisons be made?”]**
> >
> > To the best of our knowledge, at the time of submission, there was no prior work that measures planning in these settings using a quantitative diagnostic like ours.
> >
> > **[“How does VQ-VAE MI behave in settings where past work has claimed to find planning (e.g., Anthropic’s rhyming-poetry example)?”]**
> >
> > This is an excellent direction for future work. Other prior studies, including but not limited to Anthropic’s analysis, posit settings in which language models exhibit planning-like behavior. Our methodology could be applied directly to those settings, enabling an empirical comparison between previously hypothesized planning signals and the MI-based measurements produced by our method. A thorough evaluation across all such prior works would form the basis of multiple follow-up papers, and we view this as a promising extension of the present work.

---

### Official Review · Reviewer_T5EQ · 2025-10-31

**Soundness:** 3
**Presentation:** 3
**Contribution:** 3
**Rating:** 4
**Confidence:** 3

**Summary:**

The paper examines whether decoder-only language models exhibit planning behavior. It formalizes planning through three abilities: forward-looking horizon (how far future information is encoded), branch awareness (whether multiple plausible continuations are internally represented), and statefulness (how much earlier computations influence later predictions). Using an information-theoretic framework, the authors compress hidden states with a VQ-VAE and compute normalized mutual information between prefix and future representations to quantify these properties. Experiments across grammar, path-finding, and natural text tasks show that planning behavior increases with task complexity, while models remain mostly short-sighted and multi-token prediction provides only minor gains.

**Strengths:**

Please find the strengths below:
1. The paper provides a novel perspective by examining whether decoder-only language models possess planning capabilities from an information-theoretic viewpoint, making the study a fresh and meaningful contribution to the understanding of planning in LMs.
2. The use of VQ-VAE to discretize hidden states and compute normalized mutual information across layers offers a new theoretical approach for analyzing information flow, which could potentially be applied to other research problems.
3. The paper presents intuitive figures and maintains a clear logical flow, helping readers grasp the key ideas and results effectively.

**Weaknesses:**

Please find the weaknesses below:
1. Intuitively, mutual information captures only statistical correlations rather than causal relationships, and the paper does not provide theoretical justification that higher MI truly reflects stronger planning or reasoning ability; this assumption may thus be overly simplified.
2. Although the experiments cover diverse tasks, they mainly rely on small-scale models and synthetic settings, without evaluation on more recent large language models. In addition, the VQ-VAE discretization process may introduce information loss or noise, yet the paper lacks a rigorous analysis or quantification of its impact on measurement accuracy.
3. The multi-token prediction objective yields only marginal improvements, suggesting that the proposed framework, while informative, offers limited practical gains in enhancing model performance.

**Questions:**

The questions are related to the weaknesses:
1. Can the authors provide a more rigorous theoretical analysis linking mutual information to actual planning or reasoning ability, rather than relying on intuitive correlation?
2. Could future experiments include larger-scale and more capable reasoning models to test whether the observed phenomena generalize beyond small synthetic settings?
3. Can the proposed framework offer more concrete guidance or insights on how to enhance models’ reasoning or planning capabilities?
4. Could the authors further analyze how the framework applies to reasoning behaviors under Chain-of-Thought (CoT) or Tree-of-Thought (ToT) prompting settings?

---

> ### Author Response · Authors · 2025-11-20
>
> We thank the reviewer for highlighting the novelty of our work and the clarity of our presentation. Below, **we respond to all of the reviewers’ concerns and address each question** in detail.
>
> **[Connection of MI to planning, causality, and our main idea]**: We agree that mutual information (MI) measures statistical dependence, not causality. Our aim is not to claim that “higher MI causes stronger planning.” Below we explain the relation between MI and planning.
>
> Regarding why "MI reflects planning ability", we note that MI can reveal *in which part of the model is task-relevant randomness realized*, which reflects where a decision is made, hence reflecting the planning ability. To elaborate on this, consider a simple thought experiment where an LM is prompted to generate $N$ random numbers. By calculating MI between prefix hidden states and the eventual outputs, one can infer where the randomness of the random numbers is realized. If prefix states already contain high MI with final outputs (that is close to the entropy of the random numbers), it means randomness is realized at the prefix hidden states, i.e. the decision about the random numbers is already made before actual generation of the numbers, hence the model "plans ahead". On the other hand, if prefix states contain low or zero MI with final outputs, it means no randomness is realized before generation, and randomness is realized incrementally during generation, i.e. the model does not plan ahead. In this example, clearly mutual information can tell where the randomness is realized (at prefix hidden states or not), which reflects where decision is made (before generation or during generation), and hence this is an indicator of the planning ability.
>
> Secondly, **we clarify that our experiments do *not* treat absolute MI value as an indicator of planning ability**. Instead, our claim is based on comparisons of MI values. To recap, our method proceeds in three steps:
> 1. We learn high-level, discretized summaries of computations (hidden states) using our VQ-VAE.
> 2. We compute MI between these high-level variables that capture quantities of interest.
> 3. We compare MI patterns across variables, layers, and time to interpret planning behavior.
> For example, in the planning horizon analysis (Section 3.1, “Horizon of the plan”), we compare MI between the prefix hidden states and the high-level codes associated with tokens at different future offsets. If information about a token at step t+k is already present in the prefix at time t in comparable amount to the information about the immediate next token, we interpret this as forward-looking computation. The claim is comparative and structural, not causal or absolute.
>
> In the updated manuscript, we expanded the explanation of our method and the reasoning behind it.
>
>
> **[Experiment with larger models and scale]**: In addition to the scalability analysis in Appendix B.6, we now include an experiment with a larger GPT-3-type model containing over 0.3 B parameters, more than twice the size of the models used in the main paper (see Appendix D.1 for details). The results show the same qualitative trends as with smaller models, indicating that our findings generalize to larger architectures. Since the smaller model already solves the task perfectly, the larger model does not yield further improvement in planning, which aligns with our expectations. This new result also demonstrates the scalability of our framework beyond the previous experiments in Appendix B.6.
>
> **[MTP effect is marginal for performance]**: We would like to clarify that multi token prediction [1] is not a new framework we introduce, but an existing training objective that we evaluate through our information–theoretic lens. Our goal is to compare how MTP and standard next token prediction affect the planning behavior of the same model. The experiments show that while MTP provides only modest performance gains, it does not substantially alter the planning horizon. In fact, our results highlight that the task structure (CFG vs. path finding) has a stronger influence on planning than the choice between MTP and NTP.

---

> ### Author Response · Authors · 2025-11-20
>
> _**Our answers to the questions:**_
>
> **[Discretization, Accuracy of MI, and Theoretical Analysis]**: We agree that VQ-VAE introduces a lossy discretization. However, our goal is not to estimate MI between raw hidden states that include fine-grained details, but to compute MI between high-level summaries that make the measurement tractable and interpretable as discussed in the paper. Critically, **all our conclusions are comparative within the same experimental setup**. For each analysis we fix the VQ-VAE model, and we report normalized MI between the same pair types under different variable choices. For example, when comparing the MI between the summary of pre-output computations and the summary of the 1st output token vs. the 2nd output token, every component of the pipeline is held constant. This symmetry means any shift in MI can only come from the underlying interaction among the selected variables.
> We also include **a sanity check in the Appendix A.4** where the ground-truth MI is known. We show that our method preserves the ordering of MI across variables of interest, supporting our method and comparative claims using normalized MI values. Finally, even with high level representations, the plug-in MI estimator used in our study has finite-sample error. In Appendix D, **we add theoretical statement of an upper bound showing this error shrinks** rapidly with sample size, which is in the order of millions in our experiments. Therefore, the effect of finite-sample error is insignificant in our settings.
>
> **[Guidance for improving planning and relation to CoT/ToT]**: Our framework is designed to analyze planning rather than to propose new modeling techniques. Its main contribution is to offer a principled way to inspect where planning-related information appears by investigating the model’s computation from a theoretical lens. Across our three tasks we show how this yields concrete insights and we believe these insights will be useful for guiding LM research, although specific new designs or architectures are beyond the scope of this study. **Our framework can offer actionable design insights.** For example, our History in the Plan experiment can be useful for identifying where the attention window can be truncated in practice, which can enable computation-efficient LM inference. These findings illustrate that the tool can support model design decisions, even though developing new models is outside the scope of this paper.
> Furthermore, extending the framework to settings like CoT or ToT is a natural next step. Our method can in principle be applied to analyze reasoning traces in these prompting schemes by comparing how information about future steps propagates through the hidden computation during multi-step reasoning. We view this as promising future work and plan to explore it in follow-up studies.
>
> [1] Gloeckle et al., Better & Faster Large Language Models via Multi-token Prediction, 2024.
>
> We hope our responses resolve the concerns and address the questions. We would be **happy to clarify anything further**, and with this additional context and clarity, we **appreciate your consideration in raising the score**.

---

### Official Review · Reviewer_Cb1t · 2025-11-01

**Soundness:** 3
**Presentation:** 2
**Contribution:** 3
**Rating:** 6
**Confidence:** 3

**Summary:**

This paper studies planning capabilities of LLMs through an information-theoretic lens. The authors develop a framework using Vector-Quantized Variational Autoencoders (VQ-VAE) to compress the hidden states of the LLM into discrete codes. This allows them to compute mutual information (MI) estimates between different "computational blocks"/hidden states.
Using this approach, they analyze three key aspects of planning: (i) the planning horizon (how far ahead models plan before generating tokens), (ii) branch awareness (whether models internally represent alternative valid continuations), and (iii) computational history dependence (which earlier computations inform current predictions).
The framework is evaluated across synthetic grammar tasks, path-finding problems, and natural language (OpenWebText), comparing next-token prediction (NTP) versus multi-token prediction (MTP) training objectives.

**Strengths:**

- I think the idea behind the paper is quite novel: thee use of information-theoretic measures via VQ-VAE compression to study planning is a creative methodological tool

- Also, the specific focus on quantifying planning through three different characteristics is exciting. Namely, via horizon, branching, and history dimensions. This provides a more structured framework for understanding LM internal computations.

- The experimental analysis is comprehensive, including controlled synthetic tasks (CFG, path-finding) to natural language.

- The path-finding task with disjoint correct and decoy paths is particularly well-designed for testing branch awareness. I really liked the idea.

- This work addresses an important open question about whether and how LMs engage in planning-like computations. The finding that planning behavior is task-contingent has implications not only for model design, but also for training strategies.

- I believe the framework itself could also be valuable for broader interpretability research, not just planning.

**Weaknesses:**

While I really enjoyed this paper, I think it also has room for improvement:

- The main text simply says "we train a VQ-VAE", but since this is a key aspect of the method, I would expect more information about it in the main text. However, crucial aspects are only in the Appendix.

- Given the details in the appendix, the number of hidden states to store is huge, which suggests that the framework is non-trivial since the VQVAE is difficult to train.

- The experiments were carried using only GPT-3 Small models (12M-202M parameters). Findings may not generalize to larger models where emergent capabilities differ. It also becomes more difficult to apply the framework for larger models.

- The core pipeline (freeze LM -> sample block -> train VQ-VAE -> quantize -> estimate MI -> ...) is convoluted. A figure summarizing the pipeline/steps of the full framework would help a lot. Right now the reader keeps jumping between sections in the main text and appendix to figure out what is the full framework.

**Questions:**

- For each experiment, do you train one VQ-VAE per dataset/task (e.g. one for CFG prefix, one for PF prefix, one for PF paths, one for OpenWebText blocks), or per model checkpoint?

- Roughly how many sequences and optimization steps were needed to train your VAEs?

- What are the overall conclusions for NTP versus MTP?

---

> ### Author Response · Authors · 2025-11-20
>
> We thank the reviewer for recognizing the novelty of our framework, the comprehensiveness of our experimental analysis, and the strength of our experimental designs, and we are especially grateful that the reviewer genuinely enjoyed the paper. We also appreciate the reviewer’s acknowledgment of the importance of our work, answering important questions, and the potential broader impact of our framework for future interpretability research. Below, **we respond to all of the reviewer’s concerns and questions** in detail.
>
> **[More explanation on VQ-VAE / figure of pipeline]**: We thank the reviewer for highlighting this clarity issue. In the revised version, we **expanded the description** of the VQ-VAE training procedure directly in the main text, using the additional space allowed in the rebuttal phase. To further improve readability, we **added a new figure** summarizing the full pipeline, which provides a step-by-step overview of the framework. This figure and the expanded explanation appear in the updated paper (Section 2).
>
>
> **[About the size of hidden states and whether VQ-VAE is difficult to train]**:
> We thank the reviewer for raising this point. The large number of hidden states arises from the nature of the underlying task rather than from a weakness of the method. In the Planning Horizon and Branching experiments, the VQ-VAE must quantize hidden states corresponding to all prefix tokens. This results in high-dimensional vectors because the analysis requires capturing the full set of pre-output computations that may influence future decisions. The complexity reflects the difficulty of the setting we study. Importantly, our experiments show that the proposed approach can successfully handle this regime, despite the scale of the representations. For completeness, we note that the History experiment does not require quantizing such large vectors, which makes the corresponding VQ-VAE substantially smaller and easier to train.
>
> **[Experiment with larger models]**: We thank the reviewer for the insightful concern. In addition to the scalability analysis in Appendix B.6, we now include an experiment with a larger GPT-3-type model containing over 0.3 B parameters, more than twice the size of the models used in the main paper (see Appendix D.1 for details). The results show the same qualitative trends as with smaller models, indicating that our findings generalize to larger architectures. Since the smaller model already solves the task perfectly, the larger model does not yield further improvement in planning, which aligns with our expectations. This new result also demonstrates the scalability of our framework beyond the previous experiments in Appendix B.6.

---

> ### Author Response · Authors · 2025-11-20
>
> _**Our answers to the questions:**_
>
>
> **[How many VQVAEs are trained]**: Thank you for pointing this out, to make it clear we added a new section to the appendix of the paper (see Appendix A.3). Across all experiments, VQ-VAEs are trained once per dataset/task and representation type. For example, in the CFG experiments we train a hierarchical pair of VQ-VAEs (VQVAE1 and VQVAE2) to quantize prefix hidden-state blocks (for simplicity which you can think of as a single VQVAE), and a separate last-layer VQ-VAE for CFG. In the path-finding experiments, we similarly train one VQ-VAE for prefix hidden states and one for the path sequences. In the NLP experiment, we train one VQ-VAE for prefix blocks and one for last-layer representations on OpenWebText. These models are then reused across all mutual information estimations in their respective experiments. With this design, we kept the evaluation consistent and avoid conflating VQ-VAE retraining with the behavior of the underlying language models being studied.
>
> **[How many sequences and optimization steps were needed]** Again, we are thankful for reminding us about this issue, which is explained in the new section in the updated paper (see Appendix A.3). For your convenience we provide the parts in that answer that are relevant to this question: In *Planning Horizon Experiment*, three VQ-VAEs were used: the hierarchical pair (VQ-VAE 1 and VQ-VAE 2), used to quantize all prefix hidden states, were each trained for 15,000 optimization steps, yielding a total of 0.96M sequences, and an additional last-layer VQ-VAE was trained for 10,000 steps, corresponding to 0.32M sequences. The path-finding variant of planning horizon experiment used the same iteration counts and therefore consumed the same number of sequences. These amounts also applied consistently in both the multi-token prediction settings. *In Branching Experiment* , we again trained three VQ-VAEs: the hierarchical pair VQ-VAE 1 and VQ-VAE 2 are used to quantize all prefix hidden states. VQ-VAE 1 is trained for 20,000 optimization steps with 0.64M sequences, VQ-VAE 2 is trained for 15,000 steps with 0.48M sequences. The last VQ-VAE is dedicated to paths, which was trained for 25,000 steps with 0.80M sequences. Overall, the VQ-VAEs were trained with modest iteration counts, typically requiring between 0.32M and 0.96M total sequences depending on the model. In *History Experiment*, we trained two VQ-VAEs, one for prefix hidden-state block quantization and one for last-layer hidden-state quantization, each for 10,000 optimization steps, corresponding to 0.64M training sequences per model.
>
> **[Conclusions of NTP vs MTP]**: Across our experiments, we observe no major behavioral differences between next token prediction (NTP) and multi token prediction (MTP). In the horizon experiment with the CFG task, MTP leads to a very slight increase in planning capability. In the short path finding task, both objectives yield similarly effective planning behaviors, as reflected by comparable MI patterns. The clearest difference appears in the long path finding task, where MTP achieves higher task accuracy and this improvement is mirrored by higher MI values, indicating stronger information propagation across steps. Similarly, in the branch awareness experiment, the largest gap again occurs in the long path finding task, with MTP showing more branch-aware internal representations. Overall, these results suggest that MTP does not fundamentally change planning behavior, but we observe task-specific improvements.
>
> We hope our responses resolve the concerns and address the questions. We would be **happy to clarify anything further**, and with this additional context and clarity, we **appreciate your consideration in raising the score**.

---

> > ### Comment · Reviewer_Cb1t · 2025-11-27
> >
> > Thank you! The added figure and the information about the VQ-VAE training are excellent. Although the convoluted pipeline remains a weakness, I appreciate the authors' clarifications. I am increasing my score to reflect these changes.

---

### Author Response · Authors · 2025-12-01
**Summary for Area Chairs: Reviewer Discussions Prior to the OpenReview Interruption (Part 1/2)**

Dear Area Chairs,

We understand the need for the rollback after the OpenReview bug. For this submission, however, the discussion phase before the rollback was quite active and led to **explicit score increases from two reviewers** after their concerns were addressed (with post-discussion scores being 8 / 6 / 6 / 4). For the reviewers who remained at 6 and 4, we had answered their concerns in the rebuttal, but they had not yet posted follow-up comments. Below, we briefly summarize our discussion with each reviewer, including what their main concerns were, what changes we made, how they responded, and how their overall scores and sub-scores were affected.

- **Reviewer Cb1t:** The reviewer highlighted that the main text did not sufficiently explain the VQ-VAE training procedure, that the overall pipeline felt convoluted. In the revised version, we (i) expanded the description of the VQ-VAE training directly in the main text (Section 2), (ii) added a new figure summarizing the full pipeline (Fig. 1), and (iii) detailed the explanation of training VQ-VAE.  After reading the updated paper and rebuttal, the reviewer wrote that “the added figure and the information about the VQ-VAE training are excellent” and explicitly stated: “I am increasing my score to reflect these changes.” **Their rating increased from 6 → 8, and their sub-scores increased from Soundness/Presentation/Contribution = 3/2/3 to 3/3/4**.

- **Reviewer T5EQ:** The reviewer's concerns focused on (i) the conceptual link between mutual information and “planning”, (ii) the lack of evaluation on larger and more capable models, and (iii) the limited practical gains from multi-token prediction. Parts of (i) and (iii) were based on misunderstandings of our setup and claims, which we clarified in the rebuttal by explaining that we do not treat absolute MI as a causal signal, but instead use MI patterns based on comparisons across layers/offsets rather than raw MI magnitudes, and by emphasizing that the multi-token prediction loss is not our contribution but an existing method that we analyze. Concern (ii) was a valid request for stronger evidence, which we addressed by adding experiments on a larger, more capable model showing that our conclusions continue to hold in that regime. We also provided a theoretical bound on the finite-sample error of our MI estimator and discussed how the framework can suggest practical interventions. **The reviewer had not yet posted a follow-up** comment after the rebuttal, so their numeric scores remained unchanged at rating **4** with **Soundness/Presentation/Contribution = 3(good)/3(good)/3(good)**.


- **Reviewer cc54:** The reviewer initially argued, based on a misinterpretation of Xu et al. (2020), that the squared error of a linear probe (and the predictive V-information considered there) would directly provide a lower bound on Shannon mutual information, and therefore that our VQ-VAE MI estimator offered no advantage over standard linear probes. In our rebuttal, we clarified that Xu et al. introduce **predictive V-information**, not Shannon MI, and that their Proposition 1.5 (Xu et al.) relates probe loss to this quantity, which does not share key properties of Shannon MI and therefore cannot be treated as a calibrated surrogate for it. We also implemented the linear-probe baselines the reviewer suggested and showed that, in our planning-horizon and sanity-check setups, these probe-based quantities fail to recover the correct ordering in cases where our VQ-VAE MI estimator succeeds. After several rounds of discussion with new experiments and suggested writing changes, Reviewer cc54 acknowledged that their initial review contained theoretical mistakes, agreed that our method provides a useful interpretability tool that cannot simply be replaced by linear probes, and wrote: “Accordingly, I've updated my score to a 6.” Their rating changed from **4 → 6**, and sub-scores moved from **Soundness/Presentation/Contribution = 2/2/2** to **3/2/3**.

---

### Author Response · Authors · 2025-12-01
**Summary for Area Chairs: Reviewer Discussions Prior to the OpenReview Interruption (Part 2/2)**

- **Reviewer rbc9:** The reviewer already viewed the paper positively, with an **initial rating of 6**. Their main concerns were that some empirical findings might be somewhat incremental, that we do not provide a mechanistic circuit-level explanation, and that VQ-VAE-based methods may be harder to interpret and tune compared to simple probes. We clarified in our rebuttal how our framework complements mechanistic interpretability (rather than replacing it). We also addressed their remaining points by providing a more detailed comparison with probing (including new experiments), and adding a discussion on hyperparameters in the Appendix. **They had not posted a further response or changed their scores, which remained at rating 6 with Soundness/Presentation/Contribution = 3/2/2.**

With rebuttal and reviewer discussions, our score increased to **8/6/6/4** from 6/6/4/4, the initial score. The area chairs can see our scientific exchange in a completely natural way, which led to score increases. We fully abide by the ICLR code of conduct. We kindly ask that, when making the final recommendation, you consider that the reviewer's concerns were resolved and that they increased their scores accordingly. Thank you very much for your time and for carefully considering this context. If you have any remaining questions, we would be happy to clarify further within the rebuttal period.

---

### Meta-Review · Area_Chair_TAnv · 2026-01-08

**Summary:**

This paper proposes to train VQ-VAE on the top of LLM hidden states and use it to analyze the mutual information between different blocks or hidden states. Reviewers generally liked this work with a review score of 4,6,6,8 post-rebuttal (according to the reviewers' comments on improving the scores). The main concerns were: (i) whether mutual information is meaningfully linked to “planning” rather than mere correlation, and whether the term “planning” is overstated; (ii) whether the VQ-VAE pipeline is too complex / under-explained in the main text (training details, steps, reproducibility); (iii) missing or insufficient comparisons to simpler probe-based baselines; and (iv) limited scale evidence (mostly small models / synthetic tasks), plus questions about what practical actionable guidance the framework enables.
The authors did a good job on addressing most of the concerns as reflected by the score change. Overall I think this paper is interesting to understand when planning happens through a fine-grained interpretability lens. I would recommend accept of this work.

**Reviewer Concerns:**

Addressed concerns in the rebuttal:

1. Main-text clarity of the VQ-VAE training/pipeline: authors added training details in the main text/appendix;
2. “MI is correlation, not planning”: authors clarified they rely on comparative MI patterns (across layers/offsets/conditions) rather than absolute MI or causal claims, and added more explanation/intuition.
3. Probe baseline comparison: implemented probing baselines requested by a reviewer and added controlled validation experiments

Concerns still outstanding:

1. Experiments mainly rely on small-scale models and synthetic settings

**Reviewer Scores:**

Reviewer Cb1t and cc54 explicitly responded to raise the scores. From my perspective:

Reviewer rbc9 (6 → 6): Likely unchanged.
Reviewer T5EQ (4 → 4 or 6): the added larger-model experiments and expanded MI justification could may move them to borderline accept (6).

---

### Decision · Program_Chairs · 2026-01-26

Accept (Poster)